# A Robust Automated Analog Circuits Classification Involving a Graph Neural Network and a Novel Data Augmentation Strategy

**DOI:** 10.3390/s23062989

**Published:** 2023-03-09

**Authors:** Ali Deeb, Abdalrahman Ibrahim, Mohamed Salem, Joachim Pichler, Sergii Tkachov, Anjeza Karaj, Fadi Al Machot, Kyamakya Kyandoghere

**Affiliations:** 1Institute for Smart Systems Technologies, Universitaet Klagenfurt, 9020 Klagenfurt, Austria; 2Infineon Technologies Austria, 9500 Villach, Austria; 3Faculty of Science and Technology, Norwegian University of Life Sciences (NMBU), 1430 Ås, Norway

**Keywords:** analog circuits, analog subblocks, structure recognition for analog circuits, analog circuit classification, graph neural networks, data augmentation

## Abstract

Analog mixed-signal (AMS) verification is one of the essential tasks in the development process of modern systems-on-chip (SoC). Most parts of the AMS verification flow are already automated, except for stimuli generation, which has been performed manually. It is thus challenging and time-consuming. Hence, automation is a necessity. To generate stimuli, subcircuits or subblocks of a given analog circuit module should be identified/classified. However, there currently needs to be a reliable industrial tool that can automatically identify/classify analog sub-circuits (eventually in the frame of a circuit design process) or automatically classify a given analog circuit at hand. Besides verification, several other processes would profit enormously from the availability of a robust and reliable automated classification model for analog circuit modules (which may belong to different levels). This paper presents how to use a Graph Convolutional Network (GCN) model and proposes a novel data augmentation strategy to automatically classify analog circuits of a given level. Eventually, it can be upscaled or integrated within a more complex functional module (for a structure recognition of complex analog circuits), targeting the identification of subcircuits within a more complex analog circuit module. An integrated novel data augmentation technique is particularly crucial due to the harsh reality of the availability of generally only a relatively limited dataset of analog circuits’ schematics (i.e., sample architectures) in practical settings. Through a comprehensive ontology, we first introduce a graph representation framework of the circuits’ schematics, which consists of converting the circuit’s related netlists into graphs. Then, we use a robust classifier consisting of a GCN processor to determine the label corresponding to the given input analog circuit’s schematics. Furthermore, the classification performance is improved and robust by involving a novel data augmentation technique. The classification accuracy was enhanced from 48.2% to 76.6% using feature matrix augmentation, and from 72% to 92% using Dataset Augmentation by Flipping. A 100% accuracy was achieved after applying either multi-Stage augmentation or Hyperphysical Augmentation. Overall, extensive tests of the concept were developed to demonstrate high accuracy for the analog circuit’s classification endeavor. This is solid support for a future up-scaling towards an automated analog circuits’ structure detection, which is one of the prerequisites not only for the stimuli generation in the frame of analog mixed-signal verification but also for other critical endeavors related to the engineering of AMS circuits.

## 1. Introduction

Analog circuits are essential parts of systems-on-chip (SoC). A typical modern integrated circuit (IC) consists of analog, digital and mixed-signals modules. Analog circuit design and verification processes are time-consuming tasks and highly dependent on human experience [1,2]. The characteristics of analog systems are challenging, and the “always new” device models are being progressively introduced with new specifications and functionalities if compared to digital designs, which are highly automated [1]. Hence, several efforts in the related research community are focusing on automating, as much as possible, the most critical/essential tasks related to analog circuit design, verification, and layout generation. The classification or identification of circuits (later of sub-circuits) is one of those tasks.

Mixed-signal circuitry is the connection of the digital world with the “real” world. This limits the attempts to replace analog parts with digital ones. Nowadays, the so-called systems-on-chip (SoC) contain large circuits that may contain some millions of transistors, and these numbers are suspected to increase and maybe double every two years, according to Moore’s Law [3].

Consequently, dealing with this complexity of higher numbers of nano-scale transistors and coping with other technology advancements are becoming more challenging. This leads to longer times for the development process [2]. As a result, the automation of analog design, topology selection, sizing rules, layout generation, and analog/mixed signals verification are big concerns for current research.

Moreover, the recent developments in the areas of machine learning (ML) and neuro-computing (NC) do provide very recent instruments and models which can be used and optimized to realize a robust classification for complex discrete structures (with the flavor of complex graphs) such as analog circuits, which may display diverse configuration or architectures under the same label. For example, just for illustration, a current mirror may be extended by one or several banks, whereby each of those extensions still has the label of a current mirror. A particular example of recent and powerful ML models is the so-called Graph Neural Networks GCNs. In recent years, GCNs have been used to build models able to classify either analog circuits [4] or digital circuits [5]. In both cases, these circuits were converted to their equivalent text representations, or what is called in SPICE “netlists”, as they are the universal mode for circuit or schematic representation.

In [4], a GCN-based model was used to classify analog circuits under the following labels: operational transconductance amplifiers (OTA), low noise amplifiers (LNAs), mixers, and oscillators (OSC). On the other hand, in [5], a similar model was used to classify digital circuits under the following labels: Ripple-Carry Adder (RCA), Carry-Look-Ahead adder (CLA), Carry-Select Adder (CSLA), and Carry-Skip Adder (CSKA).

The comprehensive modeling of an analog circuit unto a graph model, after an appropriate related ontology formulation, and design, enables the involvement and tuning of an appropriate graph neural network architecture for ensuring a robust automated classification of analog circuits.

The core objective of this paper is to design and validate a comprehensive, robust, and reliable analog circuit classification system concept that does also cope with limited original dataset limitations w.r.t. mainly the size and eventually the missing balance amongst classes [4]. In the core, a comprehensive pipeline has to be developed which includes a graph convolutional network (GCN) model. The core GCN architecture shall be tuned and still has a couple of novel dataset augmentation techniques in order to reach maximum classification performance and robustness. To finish, a comprehensive benchmarking, at least qualitative, shall compare the quintessence of the concept developed with a selection of the most relevant related works.

The notable contributions of this paper can be summarized as follows:Assess the importance of analog circuits’ classification in different relevant application areas.A critical comprehensive review of the related current state-of-the-art. Besides a literature review, we also formulate a comprehensive ontology, starting from the netlists, for transforming a given analog circuit unto a corresponding graph model, and, further, a modeling work-flow of using machine methods (here graph neural networks) for solving the task of analog circuits’ classification or recognition.Suggest a Graph Convolutional Network (GCN) model for analog circuit classification: we develop and fine-tune a Graph Convolutional Network (GCN) architecture that may be used for structure identification in general, but here we spot the light on a “graph classification” endeavor.Suggest a novel augmentation strategy (in face of eventually limited training dataset(s) in practical settings): we develop and validate some innovative dataset augmentation concepts, which have a positive and significant boosting effect on the performance of the suggested GCN-based graph classifier model.To better underscore the novelty of the concept developed, a conceptual and qualitative comparison is performed between the concept and results of this paper and a selection of most recent competing concepts from the very recent related works.

The rest of this paper is organized as follows. Section 2 discusses the importance of analog circuit classification and formulates a related Requirements Engineering dossier. Section 3 is a comprehensive critical review of the related state-of-the-art. Furthermore, besides a comprehensive ontology for mapping an analog circuit into a graph, a general methodology for building a graph classification pipeline for a limited-size dataset is introduced in Section 4. Then, the dataset collection, its transformation, and its overall preprocessing are described in Section 5. Section 6 describes the comprehensive design of a robust GCN (graph convolutional neural network)-based classifier model. Section 7 we apply the K-Fold cross-validation (CV) on the available original limited dataset. Then Section 8 addresses the challenge due to limited-size datasets through three innovative data augmentation concepts to boost the performance of the GCN-based graph classifier. Furthermore, a series of experiments for stress-testing for the suggested pipeline and classifier is presented in Section 9 to present better the novel proposed dataset augmentation Methods. Section 10 presents the experiments’ anatomy for the different pipelines. Furthermore, for benchmarking purposes, Section 11 underscores the novelty of the concept developed in this paper through a conceptual and qualitative comparison between the concept and results of this paper and a selection of the most recent competing concepts from the very recent related works. Then, to finish, the core concluding remarks of this work are presented in Section 12.

## 2. Importance and Comprehensive Formulation of Core Requirements for a Robust Classification of Analog Circuits Module

Circuit classification is a primary stage within the automation process of most circuit design applications. Analog circuits consist of a hierarchy of building blocks at different levels. Thereby, each block of a given layer consists of lower-level building blocks, up to the lowest level of transistors and primitives [1,6,7]. Hence, classifying circuit blocks (and later recognizing circuit structures) does support the more profound knowledge of characteristics and specifications of given complex circuits (from various perspectives), especially since it has an impact on the production process of integrated circuits as follows:Consider circuit topology selection. Traditionally, the selection is made manually based on the experts’ deep knowledge of circuit design, while automating the topology for the purposes of addressing both power consumption and functionality in the face of different possible topologies from different technologies. Indeed, analog circuits classification support an automated topology selection in that it gives the knowledge needed to either select proper topologies or tune specific topology selection optimization processes [8,9,10].Consider sizing rules or the sizing process of analog circuits. It is well known that the sizing of analog circuit elements or blocks is one of the biggest challenges for designers, as circuits’ performance highly depends on the primitives’ dimensions [3]. Indeed, an automated analog circuit classification does play an essential role in guiding designers to have the suitable sizing of components in view of all possible variations that may occur and are not addressed by the topology selection in actual cases. In some cases, one may need to replace or adjust the sizes of specific blocks, lower-level transistors, or passives with other element sizes. The possible automated classification of targeted circuit groups/blocks do/shall play a considerable role in speeding up the process of elements’ sizing [1,2,11,12,13,14,15,16,17,18,19].Consider analog/mixed-signal verification. Within the context of the complicated process known as the verification of analog/mixed signals, dependable circuit classification (also known as circuit recognition) is one of the essential features that is required. Comparing a circuit’s output signal to a reference value, the expectation that the output will fall within a range of values is the last step in verifying an analog circuit that uses mixed signals. Without first understanding what this circuit symbolizes, it is impossible to make this comparison. This is why the classification of circuits or the identification of circuits is so essential [12,13,14].Consider layout generation. The ability to recognize/classify building blocks is an important capability supporting layout automation, as it provides one with a ground of optimization parameters of relevance for the layout process (e.g., expected heat generation, area, wire length, etc.) [1,2,3,6,7,8,9,10,11,20,21,22,23,24].

Since traditionally, performing structure classification does need a huge/significant human experience and a good know-how in circuit design, the intention of generalizing analog circuitry synthesis automation pushes toward the involvement of advanced techniques from the areas of Artificial Intelligence (AI) and/or Machine Learning (ML). Indeed, several AI/ML techniques have brightened the horizons for a move from the experts’ based knowledge to AI/ML-based automation. A brief review of AI/ML-based classification methods is discussed in Section 3, whereby our suggested ML model is comprehensively discussed in Section 4.

Figure 1 shows how “circuit classification” is useful for different essential use cases in the analog/mixed signals engineering business context. Finding exact solutions for analog circuit design depends on solving differential equations through simulators. This either takes a long time to find an exact solution, or one depends on presolved differential equations and thereby sacrifices a margin of accuracy. Circuit classification can help the simulator find a better selection of the presolved differential equations while ensuring relatively high accuracy, compared to a non-exact solution. It is also carried out in a relatively significantly shorter time when compared to an exact solution. Moreover, circuit classification helps in the “analog block level recognition”, which helps the block level verification and consecutively results in reducing the load of the verification engineer.

Furthermore, analog designers can benefit from knowing the type of the circuit or of the schematic by then correctly choosing the appropriate test bench to generate stimuli and perform verification and other measurements. This ensures the circuit’s functionality on the IP level and saves a lot of the designer’s load of analog/mixed signals engineering business context.

Consider a minimum “requirements engineering” dossier for a robust classifier model of analog circuits. A truly robust analog circuit classifier shall fulfill the following requirements:REQ-1: A classification accuracy (or precision) that is very high, e.g., lying in a range between 97 to more than 99 percent.REQ-2: It shall be robust w.r.t. topology/architecture variations of circuits having the same label.REQ-3: It shall be robust w.r.t. to topology/architecture variations/extension related to banks (e.g., current mirror banks of different dimensions, etc.).REQ-4: It shall be robust to variations involving different transistor technologies uniformly or mixed in the same analog circuit structure (e.g., bipolar, NMOS, PMOS, etc.) in various samples of the datasets (for both training and testing) under the same label.REQ-5: It shall be robust to practically common imperfections of the training dataset(s) (e.g., amongst others, limited dataset size, unbalanced dataset, etc.).REQ-6: The concept shall be applicable, in principle, to each of the levels of analog circuits architectures or IPs (IP refers to intellectual property packages) such as, just for illustration, level 1 (i.e. that of the basic building blocks), level 2 (i.e., that of the functional blocks), level 3 (i.e., that of the modules), etc.

It shall be indicated that the proof of concept presented and validated in this paper mostly involves the above-named levels 1 and 2. However, the developed pipeline, including the model presented in this paper, is essentially applicable and scalable to any other of the higher levels.

## 3. Comprehensive Critical Review of the Related State-of-the-Art

Verification is one of the processes to profit from a robust analog circuit classification. Indeed, the process of analog circuits verification takes a lot of time, a lot of work, and excellent human expertise (such as when it comes to building up appropriate model test benches [25], performing test simulations, and utilizing behavioral models to validate block functioning). To simplify the identification of structures and characteristics of analog circuits’ constituents of different complexity levels, analog circuits’ design and verification process require specialized solid expertise.

The majority of this information is manually implemented in production by skilled analog design engineers [6,26,27]. Numerous attempts have been made to build reliable automated analog circuit structure/component/block detection and/or identification by using different procedures and/or algorithms in light of the significant particular knowledge of human specialists that is required. First, search-oriented algorithms for subcircuit detection employing subgraph isomorphism [18], and pattern matching were described in [18,28,29,30]. These strategies were effective in identifying subcircuits, but because they use trial and error procedures, they are cumbersome and slow for big circuits.

A “probabilistic match assignment algorithm” [31,32] is combined with a “bipartite graph labeling” algorithm [33] to recognize sub-circuits using a probabilistic method involving a nonlinear function. This method computes a matrix of structure matching between devices and nets at various levels (e.g., circuits and sub-circuits). The probabilistic match assignment method was quicker than search-based approaches. Still, it had trouble keeping up with the demands of CAD software, which has a library of templates for a large number of complex circuits and necessitates more user processing to reduce computation costs [34,35]. Refer to [35] for a nonlinear graph optimization method that uses second-order terms rather than the first-order linear optimization techniques employed in [31,32] to address the issue of excessive computing complexity. The word “FROSTY” is shown in [36].

Computer software enhances earlier “pattern matching” techniques by fusing them with structure recognition algorithms on the most basic level of abstraction and creating distorted graphs for the higher level (gate level) of detected structures. In [6,37], the first library-based approaches to automatic structure recognition were presented. They are based on hierarchical libraries containing basic building blocks of transistor pairs. Improvements in the structure recognition related to a mathematical formulation and an overcoming of problems related to both ambiguity or overlapping of the single transistor are presented and discussed in [7]. In [38], a “Knowledge-based” synthesis was introduced to automatically generate layouts by means of multiple design data-based experiences and thereby proposing sub-schematics matching. For the purpose of limiting the user inputs to the structure recognition process, ref. [39] presents a machine learning technique, and ref. [40] uses a Convolutional neural network (CNN) for identification and recognition of building blocks in the process of seeking circuits layout automation without considering human inputs or experience.

In [4], a unique methodology for the classification of analog circuits into a multilevel hierarchy using a library-based primitive matching and a GCN-based machine learning method is described. The system was developed for the purpose of classifying analog circuits. The GCN-based technique is capable of handling diverse design topologies of the same sub-blocks, and this capability has been proved in a range of test cases, including two hand-crafted circuits. This approach is more scalable compared to those others that came before it, and it has been shown to be effective at classifying circuits into sub-blocks and producing circuit hierarchy trees. As the example of a switched-capacitor filter demonstrates, this may be used to guide optimization procedures such as circuit structure. Even though the GCN-based strategy does not give 100% accuracy, it is still possible to make improvements to it by employing a more varied and rich training set.

Postprocessing ensures accuracy of one hundred percent in each of the 275 test cases used for this analysis, even when just a small training set is used. Using GCN proprieties, according to [41,42], plays a key role in circuit elements classification and, Furthermore, summing up the classified elements does make higher-level new building blocks (e.g., current mirror, differential pair) to be used in various circuitry applications. Moreover, ref. [43] does present a structure recognition algorithm that is “library-free” for extracting the basic blocks of any circuit, but it does rely on some rules given to the algorithm.

Within the same scope, but for digital circuits, a GNN framework for the identification of ASIC circuit netlists was introduced in [5]. They were able to demonstrate that a GNN model has a decent performance ability when it comes to the circuit netlist recognition challenge. In order to show that GNN is capable of effectively differentiating adder circuit graphs after just two hops of graph convolution, they carried out a case-study on adder circuits. Because of this finding, it is clear that GNN can perform circuit logic discrimination by analyzing circuit structure.

After the schematic has been converted into a graph, the issue is no longer one of circuit classification but rather of graph classification. In [19], the DGCNN was presented as a unique neural network architecture for the purpose of graph classification. The DGCNN provides a number of benefits over graph neural networks that are already in use. Beforehand, it is capable of end-to-end gradient-based training since it simply takes graph data as input. This eliminates the requirement to convert graphs into tensors first, which is required by other methods. Second, it makes it possible to learn from global graph topology by sorting vertex characteristics rather than adding them all together. This is made possible by a new layer called SortPooling, which is used. In conclusion, it delivers a performance that is superior to that of previous approaches of graphs classification on a variety of benchmark datasets.

In [24], they presented a technique for sparse spatio-temporal graph-based action recognition. These graphs highlight the critical and crucial nodes and edges for GCN while excluding the unnecessary and insignificant nodes and edges. They also included a self-attention pooling method that, before to performing the pooling operation, takes into account long-term global context, node attributes, and graph topologies. Extensive tests had been done on medium-scale (UTD-MHAD, J-HMDB) and large-scale (NTU-RGBD, NTU-RGBD-120, and Kinetics-Skeleton) benchmark datasets, and the results show that this technique outperforms previous methods by a wide margin.

The core message provided by Table 1 is essentially and factually a sort of “gap analysis”. It does indeed show that none of the related works does fully satisfy the formulated requirements dossier. However, and this is the core objective, the novel concept developed and validated in this paper is supposed to fulfill all formulated requirements. This will and does underscore its innovation character w.r.t. the most relevant state-of-the-art.

## 4. General Methodology Used for Building an Analog Circuits Classifier Pipeline

In this paper, we develop a robust ML model and a comprehensive pipeline for solving the classification problem of analog circuit structures. As a proof of concept, it is applied illustratively to the level of the “basic building blocks” of analog electronic circuits. However, the concept developed can later be up-scaled to classify higher-level structures (e.g., the levels of “functional blocks”, “modules”, and further above). Here, for the proof of concept, we consider a collection of simple structures of analog circuits such as current mirror, differential pair, load, etc. as basic building blocks. At the next higher level, functional building blocks consist of more than one basic building block connected to each other through some logic.

Some examples of functional building blocks are, to name a few: level shift-er, multiplexer, comparator, low-dropout regulator or simply low-dropout regulator (LDO), filters, etc. Furthermore, at a higher level than that of the functional building blocks, is the level of modules, at which we can list entity labels such as the following ones: analog-to-digital converter (ADC), digital-to-analog converter (DAC), harmonic reduction pulse width modulation (HRPWM), a direct-current to direct-current converter (DCDC), etc.

In this paper, we develop a solution pipeline for the classification problem of analog circuits as summarized in Figure 2. The selected dataset of analog circuits, as explained in Section 4.1, consists of 5 types of current mirror banks. Each circuit sample in this dataset is first represented as a schematic built within the Cadence Virtuoso tool. Then it is converted by using the same last-named tool to a text file called SPICE “Netlist”, which is the most universal textual representation of schematics for analog circuits design or verification tools [4]. As explained in Section 4.2, the Netlist is converted into an equivalent graph model, inspired by the work in [4], by using a parser.

The parser reads/identifies the electrical elements and the so-called “nets” from the Netlist and creates an equivalent graph model where nodes and edges between these nodes are set according to the connectivity described in the Netlist. This results in an equivalent graph model that describes the original circuit well enough. The equivalent graph is represented by a matrix called the “Adjacency Matrix”. Another matrix is related to the graph; it is called the Feature Matrix. In this feature matrix, a series of properties or the nodes of the graph are described in an appropriate coding.

The feature matrix describes, for example, whether a node is either a net or an electronic element, and for electronic elements the specific type is coded. Examples of types of electronic elements are resistors, capacitors, bipolar transistors, FET transistors, etc. The two matrices described above (i.e., Adjacency Matrix, and Feature Matrix) do constitute the input of the GCN model. The GCN model does perform/realize the “classification” functionality, its output does indicate the class to which the analog circuit described by the inputs belongs; as described in Section 4.3.

Essentially, Figure 2 describes the overall system pipeline architecture developed. Here, the input analog circuit’s schematic is first exported as a netlist in Cadence virtuoso, which is a text file. Then, a parser we have developed converts the netlist file into a graph model consisting of an adjacency matrix and a feature matrix. Furthermore, the graph model is fed into a GCN model, which classifies the analog circuit sample (represented by the corresponding graph model) as one out of 5 classes described in Figure 2.

### 4.1. Proof of Concept Scenario and Dataset Issues

This work is addressing a classification problem for analog circuits. A supervised learning approach is used. The classifier will be a neural network model (here, a graph convolutional neural network) which needs an appropriately labeled dataset for comprehensive training. For the proof-of-concept, we chose to classify circuit entities belonging to the level of the so-called “basic building blocks”. However, the concept shall be upscalable to also classify higher-level circuit entities. For the selected level of the “basic building blocks”, we consider 5 classes (i.e., labels) of analog circuit entities, which are fully sufficient: CCMB, 4TCMB, WCMB, and IWCMB (see Figure 3). The selection and the creation of dataset samples representing these 5 classes do take the elements of the requirements engineering dossier (described in a previous section above) into consideration as follows (i.e., the dataset shall lay the ground for enabling a demonstration that the model developed fully satisfies those defined requirements):

Consider the Requirements REQ-2 and REQ-3: In the worst case, the dataset created contains, for some classes, up to 10 topology/architecture variations. This is done mostly through different numbers of banks for each class. For instance, Figure 3 shows three types of schematics of the same class, the cascode current mirror bank (CCMB). Figure 3a shows the CCMB with no banks and one output of the mirrored current (Net 6), and Figure 3b. shows CCMB with 1 bank and 2 outputs (Nets 6 and 8). Figure 3c shows CCMB with 9 banks and ten outputs (Nets 8, 10, 12, 14, 16, 18, 20, 22, 24, and 26). The naming of the nets is performed automatically by the Cadence Virtuoso tool after completing the schematic design. In principle, the number of banks has no limit, and it only depends on how many current replicas the designer wants/needs in his design.

Consider the Requirement REQ-4: Transistor technology variations have been considered in the dataset, for the classes that allow them. Each class of the current mirror has 10 variations due to the different numbers of banks, but this is built using a particular transistor technology. This has inspired us to repeat the same for other transistor technologies. Indeed, the nature of current mirrors is such that it allows them to be built in all 3 technologies; NMOS, PMOS, and Bipolar. This results in 30 variations of samples for each class and, thus, a total number of 150 samples is obtained for the complete dataset.

Consider the Requirement REQ-5: The dataset created is relatively small in view of usual dataset sizes in machine learning endeavors (which can reach up to several 100,000 samples). Practically, it is not possible to have hundreds of samples for each label. Moreover, the dataset created has been taken fully balanced among the classes. Although an intentionally “unbalanced” configuration is also worth consideration we have not considered that case in this work. However, the small size of the dataset is the more critical issue to tackle comprehensively.

Consider the Requirement REQ-6: The dataset created contains entities of one single level or layer of the hierarchy of analog circuits IPs. Here, the level of “basic building blocks” has been considered. Thus, after it has been proven to work for this layer, the concept and pipeline can be applied (after only a few model sizing adaptations) to any other upper layer of the hierarchy (e.g., of upper-layers: the layer of the “functional building blocks”, the layer of the “modules”, etc.). The adaptation process starts with building a dataset of higher-level blocks and then making sure the parser can correctly convert them into corresponding equivalent graphs. From that point on, the pipeline developed in this work is ready and can be used for further processing/handling for classification (data augmentation, comprehensive training, and comprehensive testing) and surely reach similarly good classification results.

In Section 5 the dataset is described in detail. Nevertheless, we are targeting 5 classes of basic building blocks, and all of them are types of current mirror banks. The first strategy for creating more samples per class or per current mirror bank type is to add more banks up to 10 per class. This resulted in a limited dataset with 50 unique samples for one transistor type. Figure 3 shows how different the samples for the same class can be, a: shows the basic structure of cascode current mirror bank (CCMB) with one output or with no banks, b: shows cascode current mirror bank of one bank, and c is a cascode current mirror bank of nine banks.

This paper also introduces three context-adapted data augmentation techniques to improve the limited size of the original dataset of only 50 samples. Thereby, the different data augmentation concepts are presented as well as the related experiments performed on the augmented training dataset for enhancing the overall classifier performance.

### 4.2. Preprocessing—The Parser

Our overall pipeline assumes that the analog circuit to be classified will be available as a SPICE circuit netlist, as Cadence Virtuoso is the tool of predilection used in analog circuits engineering processes. Cadence Virtuoso can produce a description of the circuit entity (the schematics in Cadence Virtuoso) in a text form called “netlist”. Thus, a parser is needed to interpret the netlist and build out of it a “graph model” describing the circuit entity. This graph model, the output of the parser, is the one to be processed by the GCN classifier model (see Section 4.3).

Inspired by refs. [4,18], we model each analog circuit in the dataset in form of an undirected graph G(V, E). The vertices V are split into two types. They are either transistors, one of three technologies; PMOS (CMOS), NMOS (CMOS), Bipolar, or Nets connecting two or more of these transistors. The set of edges E connect the transistors and the nets, but there is no direct connection between two transistors, nor between two nets. Hence, the graph is bipartite.

Figure 4 shows how an analog circuit, see Figure 4a, can be modeled as a bipartite graph. The four transistors M0, M1, M2 and M3, the left vertices in Figure 4b, are connected through nets represented by the right vertices in Figure 4b, and there is no direct connection between nets nor direct connections between transistors [4]. By converting the schematics to graphs changes the analog circuit classification problem into a graph classification problem. Graph classification is solved by using different approaches, such as in [16] by maintaining the support vectors at each learning step while training a classification model with SVM and a quick Weisfeiler–Lehman graph kernel.

In [17], on the basis of topological and label graph properties, a graph-classification strategy was developed. The dataset for graphs is transformed into a feature-vector dataset that can be quickly categorized with any classifier. Using a special type of graph convolution network called DGCNN, the work in [19] provides a good tool for graph classification.

The model allows end-to-end gradient-based training since it directly accepts graph data as input without first converting graphs into tensors. By sorting vertex features rather than adding them up, it makes it possible to learn from a global graph topology, and this is made possible by a brand-new SortPooling layer. Inspired by [4,5], we have built our framework to represent the circuit by a graph and then prepare this graph for the next stage of the GCN to be classified. For each graph, two matrices are created to be fed into the GCN: (a) adjacency matrix: it describes the connectivity of the graph nodes; (b) feature matrix: it contains the node features of the graph, where the types of the nodes are identified either one of three types of the transistors or just a connecting net.

In this case, the edge features are not considered to solve the classification problem. This is carried out by a parser we have developed. The main task of the created parser is to convert the netlists of the schematics to related graph models as shown in Figure 4. Furthermore, Figure 5 shows a system overview of our designed pipeline for one sample classification using GCN. Panel (a) shows an input schematic designed using Cadence Virtuoso; (b) the schematic is converted to Spice netlist (a text file representation) using the same tool Cadence Virtuoso; (c) a Spice netlist parser which converts the netlist into a graph model, and prepares the graph shown in (d) be fed into (e). The GCN is the chosen neural network model to perform the classification.

Let us recapitulate the structuring in levels of the analog circuits architecture’s hierarchical structure. This illustrates how the scaling-up of the proof-of-concept developed in the paper can be realized in a straightforward manner.

Consider Level 0: This is the level consisting of the basic electronic elements such as transistors, capacitors, etc. These are interconnected through nets in the Cadence Virtuoso schematics and thereby constitute a level-0 graph. It does contain two types of nodes: the nets and the components. There is an edge between pair of nodes, whenever there exists a direct connection between them in the schematics. It is very important to notice that in most analog/mix circuits engineering processes, any circuit or circuit element of higher levels (level-1, level-2, level-3, etc.) is always and generally provided in form of a level-0 graph only. Indeed, the netlists generated for each of those circuits, independently of the level, is always described in form of a corresponding level-0 graph model.Consider Level 1: This is the level of the basic building blocks. Their schematics provided in Cadence Virtuoso do enable the netlist-based generation of their respective level-0 graph models. Thus, a classification (involving a GCN processor model) of level-1 entities will process their respective level-0 graph models.Consider Level 2: This is the level of the functional building blocks. Their schematics provided in Cadence Virtuoso do enable the netlist-based generation of their respective level-0 graph models. Thus, a classification (involving a GCN processor model) of level-2 entities will process their respective level-0 graph models.Consider Level 3: This is the level of the functional modules. Their schematics provided in Cadence Virtuoso do enable the netlist-based generation of their respective level-0 graph models. Thus, a classification (involving a GCN processor model) of level-3 entities will process their respective level-0 graphs models.

This above-provided description of the three first levels of the hierarchy of analog/mix signals circuits IPs does clearly show why a proof-of-concept validated for level 1 is easily adaptable to handle the other higher levels. The same protocol/procedure is to be used. Indeed, the entities of all levels starting from level 1 and above have one thing in common, that is, all are expressed/described in form of a level-0 graph. Only their respective sizes vary significantly from one layer to the next direct upper layer.

### 4.3. GCN Model

As mentioned in Section 4.1, we have created more samples for each class of the current mirror banks by adding one extra bank for each new sample. This means different sizes of the equivalent graphs produced by the parser. Hence, the input graphs have different sizes in both the training and the testing datasets. The dataset creation and its appropriate transformation are better explained in Section 5. Now, to address this challenge related to various sizes of the input graph samples (representing different analog circuit samples to be classified) one of two options can be followed.

The first option is to use spectral graph convolution graph neural networks (GCN), which handle non-homogeneous (w.r.t. size) input graphs by converting them to edge lists. They are called spatial GCN [44].The second option is to use either a vanilla GCN or a spectral GCN, and to face the limitation related to the fixed size of their inputs, we (one) perform(s) zero padding to unite graph sizes to the maximum graph corresponding to the biggest schematics within the dataset [44].

Furthermore, we introduce some novel augmentation techniques in this paper, which are explained in Section 8. In Section 6 we discuss the GCN-based classifier in detail, where we convert the dataset into comprehensive graphs using a parser and the GCN to perform the graph classification. After good conditioning, the data set is split into independent train and test datasets in the following ratio: 60% for training, and 40% for testing, considering K-fold as a validation method.

## 5. Dataset Collection, Transformation, and Preprocessing

### 5.1. Dataset Collection

For the proof of concept, in this paper, a dataset of five classes, shown in Figure 6, consisting of various current mirror banks has been created. The classes considered hereby are the following ones: Cascode Current Mirror Bank (CCMB), 4 Transistor Current Mirror Bank (4TCMB), Wilson Current Mirror Bank (WCMB), Improved Wilson Current Mirror Bank (IWCMB), Wide Swing Cascode Current Mirror Bank (WSCCMB). Cadence Virtuoso was used to create the schematics of the dataset. Adding more banks allowed us to create more samples for the same class. Hence, for each one of the classes, 10 samples were created, having from zero banks to nine banks, respectively. This has resulted in a 50 schematics of level 1 circuits dataset, as shown in Table 2. The different sizes of samples within the same class are an excellent point to later stress-test the robustness of the classifier. Figure 6 shows the five classes of current mirror banks to be classified. Each one of them can be created with a different number of banks depending on how many replicas of the same current are needed in the upper-level circuit. Therefore, it is possible to add more banks on the right side of each one of the current mirror banks.

Table 2 shows the dataset created in Cadence Virtuoso. For each class of current mirror banks, 10 schematics were created by adding more banks. The first sample was created with no banks, and the last was created with nine banks. The naming convention used is sample-number-Current-Mirror-Bank-name-*i*,where iϵ[0,9] defines the number of the added banks in the current mirror bank.

### 5.2. Dataset Transformation

The netlist file is an output file of the tool Cadence Virtuoso in text form, where instances and subcircuits are defined to describe the connectivity of the electronic devices in a circuit. We will target this net connectivity of the circuit in Section 5.2 using the netlist files. The created schematics in Cadence Virtuoso can be exported to Netlists, as shown in Figure 7. The netlist text file describes the schematic in a detailed way, where every transistor name is mentioned, then its connectivity to its surrounding nets, only connected to nets as mentioned before in Section 4.2, followed by the electrical description and characterization of each transistor.

### 5.3. Dataset Preprocessing

For the purpose of converting netlists into graph models, a parser is developed, where the nodes are either primitives, such as transistors, or nets that link two or more primitives. In order to more clearly differentiate between the transistor terminals or legs, each transistor is also represented by three nodes. In the case of bipolar transistors, they are either “collector, base, and emitter”, or “gate, drain, and source” in the case of MOSFET transistors. We then create an adjacency matrix and a feature matrix for each graph using the same parser, as illustrated in Figure 8, as discussed in Section 4 [4]. In order to prepare the graphs for the GCN, the parser annotates graph components automatically. Then it creates the equivalent adjacency and the feature matrix for the annotated graph, as shown in Figure 8.

For this classification problem, we consider five unique types of elements, and we used one hot encoding to represent each of these classes, as shown in Table 3. These encodings are called feature vectors. The adjacency matrix is used to build the Feature Matrix, based on the nodes already present in a graph. The adjacency matrix determines how the feature vectors are arranged.

## 6. A Robust GCN-Based Classifier Model—Design and Validation

Inspired by convolution neural networks (CNN), Graph convolution networks (GCN) is a novel type of neural network that learns features by gradually aggregating data from the surrounding area. It multiplies input neurons with kernels for the purpose of features learning [41].

CNNs operate on neighboring pixels (structured data) and have reached great results in applications such as image classification, object detection, semantic segmentation, etc.) [45]. For instance, feedforward deep convolutional neural networks (fDCNNs) were used to classify images in biological system [46]. Moreover, in [47], an analog artificial neural network was used for low-resolution CMOS image classification. Finally, a review of GCN applications in computer vision, such as object detection, image recognition, object tracking, and instance segmentation, was shown in [48].

In contrast, GCNs clearly outperform CNNs in the case of irregular data-related endeavors such as classification [41].

Graph convolution networks (GCN) are divided into two categories:Spectral-based graph convolutional networks. Introduced by Bruna et al. [49], they are the first notable graph-based networks and do incorporate various spectral convolutional layers inspired by CNNs.Spatial-based graph convolutional networks. They were first introduced by Kipf et al. [41]. Kipf introduced a two-layer graph convolution network for semi-supervised node classification. Every node (*i*) in a graph G=(V,E) has a feature vector x(*i*) which can be combined in a feature matrix *X* of dimensions n×d where n is: the number of nodes and d is the number of input features. An adjacency matrix *A* represents the graph structure or the respective connectivity.

These two matrices, namely *A* and *X*, do sufficiently represent a graph (corresponding to a given analog circuit sample), and they are, together, the input of the GCN model. The output of the GCN model is a node-level output Z which consists of an [n × *f*] feature matrix, where *f* is the number of output features per node and n is the number of graph nodes. These outputs can go into a pooling process for representing big graphs in a better and more comprehensive way for the next layer [24].

By involving a nonlinear activation function, the neural network can be described as follows:$$H^{\left(l+1\right)}=f\left(H^{\left(l\right)},A\right)$$

As $H^{0}=X$, $H^{L}=Z$, and *L* is the number of layers. The specific models then differ only in how f(.,.) is chosen and parameterized. For large-scale graphs, some new GCN improvements have been introduced to improve the training procedures and solve the issues related to the representations learning of unseen nodes [50,51,52,53].

For structure recognition, GCN networks can have multiple layers, convolutional and pooling, which are the input of the fully connected layer for classification. In our model used in this paper, we do use a two-layers Vanilla GCN. Vanilla GCN is the simplest form of GNNs, for which its aggregation/update is an isotropic process. Isotropic process means that the characteristics of neighboring nodes are considered in the same way [54]. In addition, after every convolutional layer, a pooling layer is attached. The activation function is the rectified linear activation function (ReLU).

Motivated by [4], in our model, we have included 2 GCN layers, amongst others in view of the limited dataset; this configuration gave us the best performance. K. Kunal et al. in [4] suggested that the optimum number of convolutional layers for structure recognition shall be 2 GCN layers for reaching the best accuracy. Indeed, further increases in the number of layers cause an “oversmoothens” which drops the accuracy.

Empirically, it has been found that two-graph-convolution-layers GCN works the best, and the decision was taken depending on the grid search mechanism for the best structure. We have used different configurations and tried many simulations to decide what structure to use. In [4], it was recommended to use a two-layer structure. Nevertheless, we have explored the performance of the following structure configurations:One-layer graph-convolution GCN architecture.Two-layer graph-convolution GCN architecture.Three-layer graph-convolution GCN architecture.Four-layer graph-convolution GCN architecture.Five-layer graph-convolution GCN architecture.

Moreover, we have also tried parallel structures with 3 × 3 graph convolution layers followed by another graph convolution layer. The performance results obtained for all structures were compared, and the two-layer graph-convolution GCN architecture performed at best, as well as the parallel structure described above. Therefore, the two-layer architecture presented in Figure 9 was chosen as it does have a much-lower complexity than parallel architectures at comparable performance.

Figure 9 shows how the circuit, represented in a graph, is classified. The equivalent graph is represented in two matrices features matrix and the adjacency matrix, as mentioned before in this section. Both are used to calculate the output of the first graph convolutional layer H1 using the fast approximation of spectral convolution. H2 is the second convolution layer output calculated using the fast approximation of the convolution [4]. ReLU layers remove the negative values, and finally, the SoftMax layer votes for the detected class. Inspired by the architecture described in [4], Figure 9 shows the GCN architecture, and what we call the 2G model. This Model consists of two layers of graph convolution with ReLU activation function followed by a fully connected layer with a SoftMax activation function.

## 7. K-Fold Cross Validation

Before talking about the augmentation techniques developed and their effect impact on the performance results when the GCN model is evaluated over the test dataset, we will first apply the K-Fold cross-validation (CV) on the available original limited dataset. In general, a validation dataset, that is independent of both training and testing sets, is used to validate the GCN model. Then, the test dataset is finally used to evaluate the trained model [55]. Unfortunately, in the case of analog circuits, the luxury of a big dataset is not always available. Thus, in such a situation, a comprehensive k-fold cross-validation is useful and crucial. It allows better check of the model’s performance and evaluation comprehensively the error. The dataset is split into equal parts called folds and in the example shown in Figure 10 k is 10. In each iteration, K-1 folds are used to train the model and the other remaining 1 fold is used for validation, and then, an accuracy average is calculated.

### 7.1. List of Scenarios

In the remaining part of this section, we will present two scenarios and their results on the basic dataset before implementing any data augmentation to enhance the performance as shown in Section 8. The two scenarios are: 5 Folds, explained in Section 7.1.1, and 15 Folds, explained in Section 7.1.2.

#### 7.1.1. The 5 Folds Case

In this scenario, the dataset is split into 4/5 for the training set and 1/5 for the test set at each fold. In other words, 1/5 of the dataset is used for testing and the rest is used for training. Table 4 shows a confusion matrix drawn for each one of the five folds, and they show 100% accuracy for fold 4 and 90% accuracy for fold 1. Hence, the GCN model’s overall performance can be evaluated by the mean accuracy of the k-fold cross-validation for five folds, which is 98%.

#### 7.1.2. The 15 Folds Case

By training the GCN model for 15 folds of the dataset, it shows that the mean accuracy of the k-fold cross-validation, in this case, is 100%. In each fold, 1/15 of the dataset is used for testing and the rest 14/15 is used for training independently. Table 5 shows that for each fold a confusion matrix is drawn, and all confusion matrices show 100% classification accuracy.

## 8. Addressing the Challenge of Limited Datasets through Innovative Data Augmentation Concepts for Better Training the GCN Classifier

### 8.1. Related Works

Graph Data Augmentation (GDA) creates new graph data objects via modification or creation. However, the data objects in graph machine learning are not always independent and identically distributed, i.e., non-i.i.d, since graphs contain linked data rather than pictures or words. As a result, GDA approaches alter the complete dataset (graph) for node-level and edge-level tasks rather than just a few data elements (nodes or edges). We describe four kinds of GDA activities based on the changed or newly formed graph data components [57]. GDA techniques can be classified as follows:

#### 8.1.1. Node Augmentations

The GDA procedures known as “Node Augmentations” add or delete nodes from the graph. For instance, mixup-based approaches [57] combined two existing nodes to produce new nodes. The “DropNode” technique was suggested by Feng et al. [57] and deleted nodes by hiding the characteristics of chosen nodes.

#### 8.1.2. Edge Augmentations

Edge augmentations are GDA procedures that change the graph’s connectivity by adding or deleting edges. The alterations may either be deterministic (such as GDC and GAug-M [57], which both altered the graph structure and utilized the changed graph for training/inferencing) or stochastic (e.g., Rong et al. [57] proposed to drop edges during each training epoch randomly).

#### 8.1.3. Feature Augmentations

The GDA procedures that alter or produce raw node characteristics are known as feature augmentations. For instance, FLAG [58] enhanced node features using adversarial perturbations based on gradients as opposed to [59]’s attribute-masking technique that randomly removed node characteristics.

#### 8.1.4. Subgraph Augmentation

The GDA techniques known as “subgraph augmentation” works at the graph level and include things such as clipping out subgraphs and creating new graphs. The subgraph augmentation procedures are often utilized for graph-level activities since they typically impact several subgraph nodes. For instance, ifMixup [60] combined two graphs to form a new graph, whereas JOAO [61] employed subgraph cropping.

In [57], and in [62,63,64,65], GDA techniques for supervised learning are categorized as node-level, graph-level, and edge-level tasks. In our classification problem, we focused on graph-level augmentation techniques. Specific augmentation techniques in computer vision CV and natural language processing NLP may be applied to graph data when data items are independent graphs. For instance, GraphCrop [66] crops a sub-graph from each of the provided graph objects, much like picture cropping. To preserve the topological properties of the original graphs, GraphCrop uses a node-centric technique based on graph diffusion.

Motifs are used by M-Evolve [67] to enhance the graph data. Prior to adding or removing edges inside the chosen motifs, M-Evolve first identifies and chooses the target motif in the graph using the Resource Allocation index. Similarly to this, MoCL [68] enhances the molecular graphs on the substructures such functional groups, by using biomedical domain knowledge. Each molecular graph has a substructure, which MoCL chooses and replaces with a different substructure [57].

There were also other Mixup algorithms put up for graph categorization. For instance, the Graph Mixup method stated before [69] also functions for graph classification. The latent representations of the two graphs are mixed in Graph Mixup. In contrast, ifMixup [60] applies Mixup directly to the graph data instead of the latent space. Due to the irregularity of the two graphs and the fact that the nodes from the two graphs are not aligned, ifMixup randomly allocates indices to each graph’s nodes and matches the nodes in accordance with the indices.

Graph Transparent [70] blends graph in data space similarly to ifMixup. Graph Transparent preserves the local structure by using substructures as mixing units instead of ifMixup, which randomly matches nodes during mixing. In order to choose one relevant substructure from each network, Graph Transparent uses node salience information, which includes attribute masking, subgraph sampling, and subgraph sampling that randomly adds or deletes edges, node attributes, and subgraph attributes.

These various above-listed augmentation methods alter the graph’s structure, making it impossible for us to utilize them to solve our circuit classification problem since they alter the connection of the circuits. As a result, we’ve created a few alternative augmentation methods presented in this paper that are more focused on the subject at hand—classifying schematics linked to analog circuits. The dataset we will utilize for this classification problem has a limited amount of original samples for each class, which in our case are all various types of current mirror banks. The primary objective here is to increase the training dataset in order to improve the GCN’s overall performance.

## 9. Proposed Dataset Augmentation Method

Since we are using a spectral-based GCN, the graph related to each analog circuit sample is represented by an adjacency matrix and a feature matrix. This section shows how our few new specific augmentation techniques will apply and modify either one or the other or both of these features of the nodes X. We have developed and applied two specific augmentation techniques. The first form of augmentation we have applied is what we call “physical or electrical augmentation”, explained in Section 9.1.

The second form of augmentation is what we call “numerical augmentation”. This numerical augmentation, explained in Section 9.2, consists essentially of adding a small random number α to each of the nodes features.

### 9.1. Electrical Augmentation

Here we provide details of the so-called “electrical augmentation”. The dataset used consists of five classes of current mirror banks as shown in Figure 6. Those five classes are Cascode Current Mirror Bank (CCMB), 4 Transistor Current Mirror Bank (4TCMB), Wilson Current Mirror Bank (WCMB), Improved Wilson Current Mirror Bank (IWCMB), Wide Swing Cascode Current Mirror Bank (WSCCMB).

By changing the transistor type, the initial/original dataset of 50 samples of these current mirror banks can be correspondingly multiplied or duplicated. Thus, these original 50 samples are duplicated for each transistor type, which means that the schematics built in Cadance Vertuoso can now be based on three different transistor types:NMOS: 50 Schematics (original dataset).PMOS: 50 Schematics (first duplication).bipolar: 50 Schematics (second duplication).

This above described “electrical augmentation” does therefore result into 150 samples of an augmented dataset (still containing 5 classes). It is then used to better train and test the GCN model. The dataset of course, as usual, is split into a test dataset and a training dataset, which are independent and this will be further discussed in this Section.

### 9.2. Numerical Augmentation

Here, we provide details of the so-called “numerical augmentation”. Since we represent each graph with two matrices, an adjacency matrix and a feature matrix, we can manipulate these two matrices in order to increase the number of samples in the training dataset.

#### 9.2.1. Feature Matrix Augmentation

A feature matrix consists of one-hot-encoding vector for each of the nodes within the graph. Hence, creating a replica of each graph by adding a small random number α, e.g., within the interval [0.001, 0.01], to the feature matrix elements, will correspondingly duplicate/multiply the training dataset size. Figure 11 shows different scenarios of training with parts of the training dataset TDS and compensate the missing part with augmented data to end up with the same size of the training dataset. Figure 11a is a representation of the 100% of the training dataset of 100 graphs. Figure 11b is a representation of TDS/2, a balanced half of the training dataset (50 graphs), and augmented data of 50 graphs to compensate for the missing part of the cutdown data and so to end up with 100 graphs for training. Figure 11c shows TDS/4 or 25% of the training dataset and 75% of augmented data. Figure 11d shows TDS/10 or 10% of the training dataset and 90% augmented data. In all cases, the test dataset is fixed and it has 50 independent graphs.

We performed four augmentation experiments, each one including seven scenarios:SCENARIO 1: Using the whole training dataset TDS (i.e., 100%)SCENARIO 2: Reduced training data (a selection) to a portion of the original data; specifically to ½ of the original training dataset.SCENARIO 3: Reduced training data (a selection) to a portion of the original data; specifically to ¼ of the original training dataset.SCENARIO 4: Reduced training data (a selection) to a portion of the original data; specifically to 1/10 of the original training dataset.SCENARIO 5: Augment to compensate the missing data samples in scenario 2; shown in Figure 11b.SCENARIO 6: Augment to compensate the missing data samples in scenario 3; shown in Figure 11c.SCENARIO 7: Augment to compensate the missing data samples in scenario 4; shown in Figure 11d.

For each one of these scenarios, 5 trials or simulations were conducted and the mean accuracy was reported.

Experiment-1: Add the same random number α from the interval [0.001, 0.01] to all non-zeros elements in the feature matrix X to construct X2 as a new feature matrix of the same graph. Table 6 shows a performance comparison of different scenarios, numbered as explained above, where we take a part of the training set for training and we keep the same test dataset for testing. The first two lines show the mean accuracy for each dataset case, where we take for training half of the dataset, a quarter, then a tenth, and then test with the same test dataset of 50 graphs. As expected, the mean accuracy drops when the model has fewer data to train. The second two lines show the compensation we did with the augmented data as explained in Figure 11. The accuracy is better compared to the first two lines, but it is still far from the accuracy where we use the whole training dataset TDS for training. The division sign ‘/’ is used to refer to taking part of the dataset. For example, TDS/2 means half of the dataset TDS. The asterisk sign ‘*’ refers to the augmentation, and the number after it refers to the scale of augmentation, or how many replicas we do using the augmentation technique described in each experiment.

**Table 6 sensors-23-02989-t006:** Experiment-1: Performance comparison; TDS is the original dataset where 100 graphs are for training and 50 graphs for testing. The Numerical Augmentation ratio is Dataset × Augmentation-scale.

Dataset	1. TDS	2. TDS/2	3. TDS/4	4. TDS/10
**Mean Accuracy reached after training with the reduced datasets**	90.50 ± 0.26	48.20 ± 0.52	40.05 ± 0.74	32.20 ± 0.54
**Augmented Dataset**	1. TDS	5. (TDS/2)*2	6. (TDS/4)*4	7.(TDS/10)*10
**Mean Accuracy (in%) reached after testing the model training by the augmented datasets**	90.50 ± 0.26	50.54 ± 0.15	42.86 ± 0.25	30.19 ± 0.62

Experiment-2: Add the same random number α from the interval [0.001, 0.01] to all elements in the feature matrix X to construct X2. Table 7 is similar to Table 6 in terms of scenarios implemented and the numbers of scenarios were removed for simplicity. Table 7 shows that the accuracy (see Table 7) is worse than the first experiment (see Table 6). This means that adding a small random number to the zeros within the feature matrix does result in a some confusion to the model.

**Table 7 sensors-23-02989-t007:** Experiment-2: Performance comparison; TDS is the original dataset where 100 graphs are for training and 50 graphs for testing. The Numerical Augmentation ratio is Dataset × Augmentation-scale.

Dataset	1. TDS	2. TDS/2	3. TDS/4	4. TDS/10
**Mean Accuracy reached after training with the reduced datasets**	90.50 ± 0.26	48.20 ± 0.52	40.05 ± 0.74	32.20 ± 0.54
**Augmented Dataset**	1. TDS	5. (TDS/2)*2	6. (TDS/4)*4	7. (TDS/10)*10
**Mean Accuracy (in%) reached after testing the model training by the augmented datasets**	90.50 ± 0.26	37.61 ± 0.84	28.35 ± 0.55	22.70 ± 0.50

Experiment-3: Add a random α from the interval [0.001, 0.01] to each non-zero element (the random number will be different for each element of the feature matrix) in the feature matrix X to construct X2. Table 8 shows a big enhancement compared to the first two experiments. The new augmented feature matrices have the same structure as the original ones with the small tuning of the non-zero elements, and apparently, this helped the model cope with the new samples of the test dataset.

**Table 8 sensors-23-02989-t008:** Experiment-3: Performance comparison; TDS is the original dataset where 100 graphs are for training and 50 graphs for testing. The Numerical Augmentation ratio is Dataset × Augmentation-scale.

Dataset	1. TDS	2. TDS/2	3. TDS/4	4. TDS/10
**Mean Accuracy reached after training with the reduced datasets**	90.50 ± 0.26	48.20 ± 0.52	40.05 ± 0.74	32.20 ± 0.54
**Augmented Dataset**	1. TDS	5. (TDS/2)*2	6. (TDS/4)*4	7. (TDS/10)*10
**Mean Accuracy (in%) reached after testing the model training by the augmented datasets**	90.50 ± 0.26	76.60 ± 0.32	58.75 ± 0.34	40.80 ± 0.50

Experiment-4: Add a random α from the interval [0.001, 0.01] to each element (the random number will be different for each element of the feature matrix) in the feature matrix X to construct X2. Table 9 shows a decrease in the performance of the model for the same reason in experiment 2, which is adding random numbers to the zero elements of the feature matrix.

**Table 9 sensors-23-02989-t009:** Experiment-3: Performance comparison; TDS is the original dataset where 100 graphs are for training and 50 graphs for testing. The Numerical Augmentation ratio is Dataset × Augmentation-scale.

Dataset	1. TDS	2. TDS/2	3. TDS/4	4. TDS/10
**Mean Accuracy reached after training with the reduced datasets**	90.50 ± 0.26	48.20 ± 0.52	40.05 ± 0.74	32.20 ± 0.54
**Augmented Dataset**	1. TDS	5. (TDS/2)*2	6. (TDS/4)*4	7. (TDS/10)*10
**Mean Accuracy (in%) reached after testing the model training by the augmented datasets**	90.50 ± 0.26	40.33 ± 0.82	32.22 ± 0.37	24.70 ± 0.60

We have repeated the four experiments with a different range for a random α2, by taking the new range of [0.0001, 0.001]. For this different range for α2, once again, the results obtained for Experiment-3 (where α2 is added to the non-zero elements in the feature matrix), were the best ones, this similar to the previous set of experiments (where α2 was in a different interval). The results of this last-named repetition of the four experiments are presented in Table 10. Experiment 1 showed better accuracy than experiments 2, and 3 because we added the same α2 to the non-zero elements of each feature matrix, but still Experiment 3 was better since we add a random α2 to each element in each feature matrix.

**Table 10 sensors-23-02989-t010:** The four previously described experiments were repeated for a random α2, sampled from the new range of [0.0001, 0.001]. Once again experiment-3, where α2 is added to the non-zero elements in the feature matrix, showed the best results.

**Experiment 1**				
Dataset	TDS	TDS/2	TDS/4	TDS/10
Mean Accuracy	90.50 ± 0.26	48.20 ± 0.52	40.05 ± 0.74	32.20 ± 0.54
Augmented Dataset	TDS	(TDS/2)*2	(TDS/4)*4	(TDS/10)*10
Mean Accuracy	90.50 ± 0.26	58.50 ± 0.72	46.32 ± 0.38	32.56 ± 0.43
**Experiment 2**				
Dataset	TDS	TDS/2	TDS/4	TDS/10
Mean Accuracy	90.50 ± 0.26	48.20 ± 0.52	40.05 ± 0.74	32.20 ± 0.54
Augmented Dataset	TDS	(TDS/2)*2	(TDS/4)*4	(TDS/10)*10
Mean Accuracy	90.50 ± 0.26	36.54 ± 0.15	26.75 ± 0.78	20.95 ± 0.84
**Experiment 3**				
Dataset	TDS	TDS/2	TDS/4	TDS/10
Mean Accuracy	90.50 ± 0.26	48.20 ± 0.52	40.05 ± 0.74	32.20 ± 0.54
Augmented Dataset	TDS	(TDS/2)*2	(TDS/4)*4	(TDS/10)*10
Mean Accuracy	90.50 ± 0.26	78.60 ± 0.45	60.46 ± 0.55	44.85 ± 0.44
**Experiment 4**				
Dataset	TDS	TDS/2	TDS/4	TDS/10
Mean Accuracy	90.50 ± 0.26	48.20 ± 0.52	40.05 ± 0.74	32.20 ± 0.54
Augmented Dataset	TDS	(TDS/2)*2	(TDS/4)*4	(TDS/10)*10
Mean Accuracy	90.50 ± 0.26	37.88 ± 0.46	25.45 ± 0.87	21.44 ± 0.35

Apparently, this randomness helped the model cope with the new samples in the test dataset, despite the fact that the added α2 is relatively small compared to 1 the highest value in the feature vector defined. Experiments 2 where we add the same α2 to all elements in the feature matrices of the augmented data, and experiments 4 where we add different random α2 to all elements as well, resulted in bad accuracy due to the confusion introduced to the model by the absence of all zeros in each feature matrix in the augmented dataset.

#### 9.2.2. Dataset Augmentation by Flipping

Each graph is represented by two matrices: an adjacency matrix A and a feature matrix X. It is further stored in the dataset by putting both together: [A|X]. In this type of augmentation (namely: augmentation by flipping), we create a sparse matrix much larger than the adjacency matrix, which we store in this sparse matrix in one corner, and then we flip or shift it to another corner to create a new augmented sample of the same original one, whilst the feature matrix X is fixed for all augmented samples. Figure 12 shows how the dense part is flipped systematically for each adjacency matrix in the training set, and the original feature vector is fixed for all augmented samples. Then we extend the training dataset with the new augmented adjacency matrix concatenated with the original feature vector as follows [Adj Augmented | X Original]. Figure 13 shows augmentation by flipping. The original test dataset was kept the same. On the other hand, the training dataset was augmented from 100 graphs up to 32,000 graphs through 5 stages.

Figure 14 shows the performance enhancement by increasing the number of training samples in the training dataset by flipping. We have augmented the training dataset while keeping the testing dataset fixed and independent. This has resulted in performance enhancement. Figure 14 shows two scenarios:Without Alpha; applying the flipping augmentation on the original feature vector, and checking the performance enhancement by increasing the number of samples.With Alpha; we multiplied each feature vector by a unique alpha sampled uniformly from [0.0001, 0.001].

#### 9.2.3. Multi-Stage Augmentation

According to [71], using various data augmentations, including random cropping, color distortion, and Gaussian blur, is crucial for visual tasks. The authors demonstrated that learning effective representations requires more than a single modification.

We suggest here the designing of a multi-stage augmentation in order to fully take use of the generalizing capacity and improve the variety and quality of adversarial perturbations. We use the so-called “multi-stage augmentation”. In order to combine the benefits of all previously described augmentation schemes, electrical, numerical, and flipping augmentations, we have created a further augmentation scheme that we call “multi-stage augmentation”.

By applying the electrical augmentation, as explained in Section 9.1, we augmented the basic dataset of 50 graphs to 150 graphs. We split this dataset to 100 graphs for training and 50 for testing. We augment the 100 graphs of the training set by performing flipping augmentation, described in Section 9.2.2, to double the training dataset size to 200 graphs, then we perform a “numerical augmentation” on that resulted dataset to multiply its size by 6, which means we have after these two stages 1200 graphs for training, as shown in Figure 15.

We have compared two cases. In the first one, we used the resulting dataset of the electrical augmentation, which is 150 graphs. Of these, 100 graphs are for training and 50 for testing. In the second case, after applying the flipping augmentation, a numerical augmentation is applied on the same 100 graphs of training to obtain a new set of 1200 graphs for training, but we keep the same 50 graphs for testing. The chosen GCN model (i.e., a so-called 2G GCN model, shown in Figure 9) was trained and a classification accuracy of 94% was achieved in average, and a maximum accuracy of 98% accuracy, as shown in Table 11a.

The GCN model (i.e., the 2G GCN model) was also trained again. The classification performance results obtained after the multi-stage augmentation are presented in Table 11b. Notice that the test dataset of 50 samples was used for testing in both cases. The results presented in Table 11 show that the classification performance over the test dataset is very high, almost 100%. This means that this multi-stage augmentation method does allow us to achieve a clear 100% accuracy.

#### 9.2.4. Hyperphysical Augmentation

Here we provide details of the so-called “hyperphysical augmentation”. Indeed, the simple physical (or electrical) augmentation allowed us already (see above) to create extra 100 new samples (in additionally to the 50 original ones), which are still equivalent/corresponding to real schematics since the current mirror banks can be built using the three technologies which are NMOS, PMOS, and bipolar. Table 12 shows the feature vector length; in this case, it is 4 for representing the one hot-encoding of the four possible node types: three transistor types and one net type.

This new extended structure of the feature matrix has motivated us to go one step further in the face of the problem related to limited datasets in this case of schematics. We have decided to create another extended feature matrix style that leads to a much bigger dataset through creating/using/defining 14 features described in Table 13 as follows; see also Table 14. Here, we essentially introduce a series of fictive features to reach 14 features.

The here (above) described “hyperphysical augmentation” has resulted in building a new dataset consisting of 600 graphs. Hereby, each node in a graph sample can have/take a feature out of 14 features shown in Table 14, where it could be an electrical element out of either 13 unique ones or a net.

Since the GCN model we are using is a spectral graph convolution one, the input graph size must be fixed and unique. For this, it shall correspond to the biggest circuit within the dataset. Thus, for all circuits (graphs) smaller than the biggest one, zero padding is applied to all samples to make sure that the size matches the biggest graph size of 47 nodes. This biggest size is equivalent to one of the biggest schematics, which is a current mirror bank of nine banks. Table 15 shows the new dataset after performing hyperphysical augmentation on the dataset of 150 graphs and thereby increasing the number of samples to 600. The augmentation resulted in 120 samples per class.

To evaluate the impact of the hyperphysical augmentation, the dataset (of a total of 600 samples) was split into 400 graphs for training, and 200 for testing. Each class is equally represented in the tests and train data, whereby for each class 80 graphs were used for training and 40 graphs for testing, randomly sampled at each experiment in a balanced manner. We have trained the GCN model (i.e., the 2G GCN model) using this novel hyperphysically augmented dataset for 1000 epochs and 10 trials. As result, a mean accuracy of 100% was achieved as shown in Table 16.

#### 9.2.5. Numerical Augmentation of the Hyperphysically Augmented Dataset

We now assess how far an initial dataset obtained by “hyperphysical augmentation” can be further augmented by using a numerical augmentation. We repeat the numerical augmentation experiment by adding a random α value from within the range [0.0001,0.001] to the non-zero elements in the feature matrix. We work on the dataset of 600 samples obtained through the new hyperphysical augmentation. Hereby, 400 samples are taken for the training dataset and the remaining 200 samples are the test dataset. The reference size of the training dataset, i.e., 100%, corresponds to 400 samples. In a series of scenarios, the training dataset is reduced and consecutively, we see how far the augmentation (numerical augmentation) performed on the reduced training dataset can then increase the classification performance progressively.

The first scenario uses only half of the training set and reports the corresponding reduced accuracy. Then we compensate the other half (of non-used samples of the reference training set) by a numerical augmentation and then we report the accuracy again. We repeat this further for respectively a quarter and a tenth of the reference training dataset.

We have performed five trials for each experiment, by involving our GCN model and do train for 1000 epochs. The mean accuracy values reached for the different scenarios are reported in Table 17.

Table 17 shows the importance of applying more than one augmentation technique. The first column from the left shows the result of the hyperphysical augmented dataset. The dataset described in Section 9.2.4 consists of 600 graphs split into 400 for training and 200 for testing. The second column shows first (TDS/2) the case where we trained the model with half of the training set i.e., 200 graphs and tested with the same test set of 200 graphs to achieve 98.20 ± 0.32% accuracy. This decrease of accuracy resulted from decreasing the number of training samples. Then, (TDS/2)*2 means that we have augmented the missing half by applying the numerical augmentation method and then test with the independent test dataset of 200 graphs to achieve 100% accuracy of classification.

This result is shown in Table 18 in order to prove the importance of applying more than one augmentation technique, especially on a limited dataset. The third column shows when we train the GCN model using a quarter of the training dataset TDS/4 and test with the same test set of 200 graphs, the test classification accuracy was 69.47 ± 0.57% which was improved using the numerical augmentation as above to 92.15 ± 0.46%, which is a good improvement but not sufficient to our requirements engineering. Of course, one can augment the training dataset (TDS/4) more using numerical augmentation trying to enhance the accuracy, but unfortunately, for this case, the results were not improving for more augmented data. We repeated the same experiment for a tenth of the training dataset TDS/10 and got 49.85 ± 0.34 % and 61.55 ± 0.50 after augmenting numerically ten times as shown in the last column.

As mentioned in the last paragraph, we have conducted 10 trials for this experiment while involving our GCN model (i.e., the 2G model). in each trial the GCN model is trained for 1000 epochs. Table 18 shows the result of the numerically augmented dataset (TDS/2)*2, the mean accuracy is 100% and the confusion matrix shows that all 200 test samples in the dataset were classified correctly.

## 10. Experiments Anatomy for the Different Pipelines

We have built a comprehensive pipeline to be used for our different classification experiments. An input schematic shall be classified into one out of five classes. The overall and essential anatomy of these experiments, as shown in Figure is described through the following steps:The first pipeline was designed to train and test the GCN model using a limited dataset of 150 graphs. The dataset was split into 100 graphs for training and 50 for testing. The classes were balanced, since the samples have different sizes.In order to enhance the preliminary classification accuracy, numerical augmentations were applied. As explained in Section 9.2, four experiments were made to add small value α to the elements of the feature matrix. The third experiment of adding a random α to the non-zero elements of the feature matrix gave the best results among the four experiments.Enhanced Pipeline: we have enhanced the pipeline to have more reliable and consistent results by performing the following changes:Enhanced Architecture: improving the hidden layers’ parameters, the number of GCN layers, and the number of fully connected layers.Designing the custom learning scheduler: we have chosen an exponential decay: It divides the learning rate for every epoch by the same percentage (%). This indicates that the learning rate will gradually decline over time, spending more epochs with a lower value but never reaching zero.Extensive hyperparameter tuning: fix all parameters and optimize one to check the impact of tuning the respective hyperparameters on the performance results.Small dataset with enhanced pipeline: only 1/3 of training data were shown to the model to check the enhancement achieved by applying the various augmentation techniques on the small training dataset.

The overall setup of the classifier pipeline is configured as follows (see Figure 16):Balanced dataset sampling for training and testing.Randomized shuffle each trial.Reporting mean accuracy of 10 trials for each experiment.Reporting a confusion matrix to analyze the analysis of the results.

## 11. Qualitative and Conceptual Comparison of the Concept Presented in This Paper with a Selection of the Most Relevant Related Works, Especially from the Perspective of the “Comprehensive Requirements Dossier” at Stake

The engineering requirements addressed in Section 2 were totally satisfied by the final best pipeline that has been progressively constructed and presented in this work and by the GCN model that supported it. We were able to achieve a classification accuracy of 100%. Therefore, the conditions for very accuracy were fully satisfied.

In Table 19 we comprehensively summarize how far all elements of the comprehensive requirements dossier have been fully satisfied and thereby also put the finger on the respective limitations of the corresponding related most relevant works from the literature.

## 12. Conclusions

This paper has developed and validated a comprehensive method for robustly classifying schematics or analog electronic circuits by using a graph convolutional network GCN-based system. This innovative approach considers both the structure of the schematics and their capacity to be represented as graphs. After constructing a reliable GCN-based classifier, we have focused on the sensitive issue of a limited dataset size of the schematics. As a result, various forms of dataset augmentations have been presented and evaluated on both the schematic level and the graph level to improve the overall system performance.

For graph-related machine learning, we have offered/suggested a complete and comprehensive set of data augmentation approaches. We could discuss the rationale for developing our own methods, offer contemporary GDA approaches based on their methodology, and comprehensively evaluate the different known GDA techniques. We have demonstrated that the challenges related to limited dataset size could be significantly tackled and improved.

Our extensive experiments convincingly demonstrate that our overall concept, including the neural networks model and the augmentation techniques, can very robustly/reliably classify analog circuit schematics correctly out of five classes. This good performance can be upscaled towards a library-based analog structure classification for a series of endeavors such as simulator performance tuning, analog block-level verification, and IP-level verification.

Furthermore, in the beginning, this study evaluated the significance of the classification of analog circuits in various application domains. A critical and complete analysis of the current state-of-the-art in the relevant field was also conducted. In addition to conducting a literature review, we have formulated a comprehensive ontology, beginning with the netlists to transform a given analog circuit into a corresponding graph model. Additionally, we have developed and validated a global modeling workflow involving graph neural networks to solve the problem of analog circuits’ classification or recognition. Furthermore, we have suggested and comprehensively evaluated various novel augmentation strategies (in the face of the potentially limited training dataset(s) in practical settings). We could develop and validate some innovative dataset augmentation concepts, which positively boost the performance of the suggested GCN-based graph classifier model.

To better underscore the importance of this study, a conceptual and qualitative comparison has been carried out between the overall concept and the results of this paper and a selection of the most recent competing concepts from the very recent related works. This was done to emphasize better the novel nature of the global concept developed and validated in this paper. Overall, this paper is of high interest and significant practical importance as a reference for academics and practitioners working on analog circuit design automation issues or on various graphs in learning problems and approaches.

## Figures and Tables

**Figure 1 sensors-23-02989-f001:**
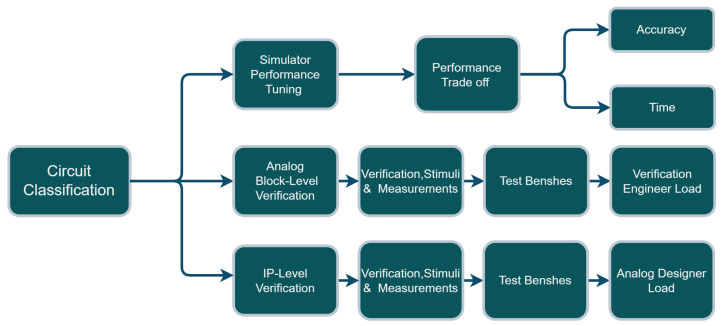
Circuit classification use cases: Simulator performance tuning, analog block-level verification, and IP-level verification.

**Figure 2 sensors-23-02989-f002:**
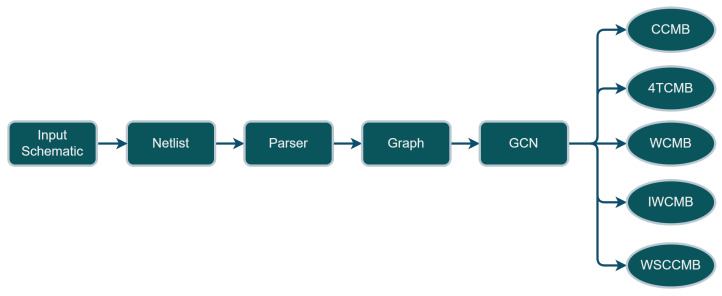
Overall system PIPELINE of the developed Analog Circuit Classifier. An input schematic is converted to a graph model through a parser we have developed. Then, the equivalent graph model is fed into the GCN to be classified into one of the following classes: Cascode Current Mirror Bank (CCMB), 4 Transistor Current Mirror Bank (4TCMB), Wilson Current Mirror Bank (WCMB), Improved Wilson Current Mirror Bank (IWCMB), Wide Swing Cascode Current Mirror Bank (WSCCMB).

**Figure 3 sensors-23-02989-f003:**
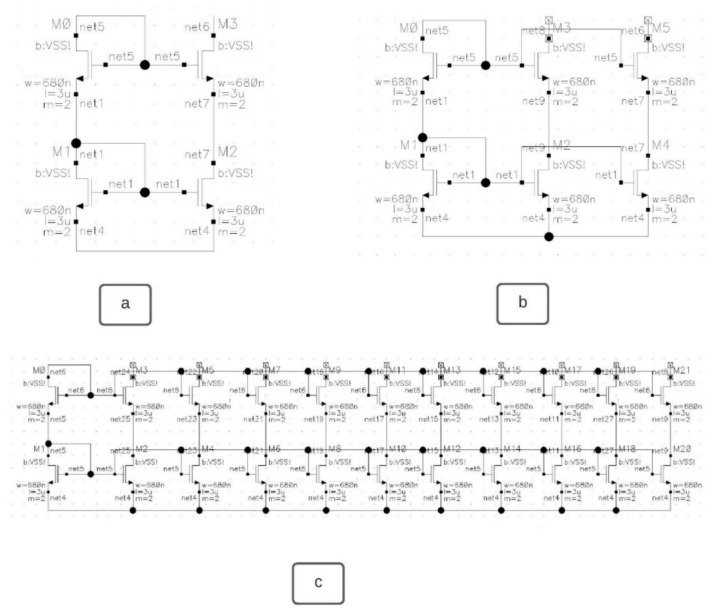
Example schematics samples of the same class “cascode current mirror bank (CCMB”). Different configurations are presented here: (**a**) 0 bank; (**b**) 1 bank; and (**c**) 9 banks. For each other class (in total we have 5 classes) within the dataset, there are similarly 10 topology/architecture variations.

**Figure 4 sensors-23-02989-f004:**
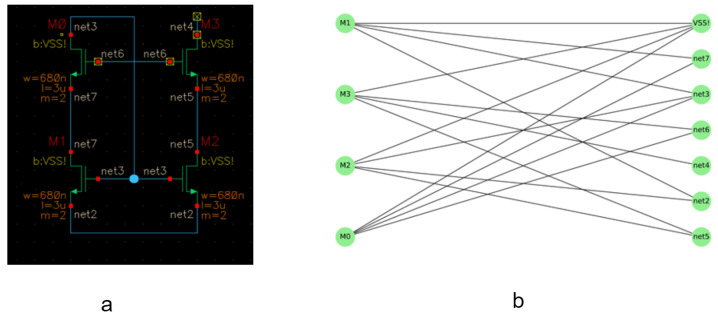
(**a**) An NMOS current mirror of four transistors. (**b**) its representation as a bipartite graph.

**Figure 5 sensors-23-02989-f005:**
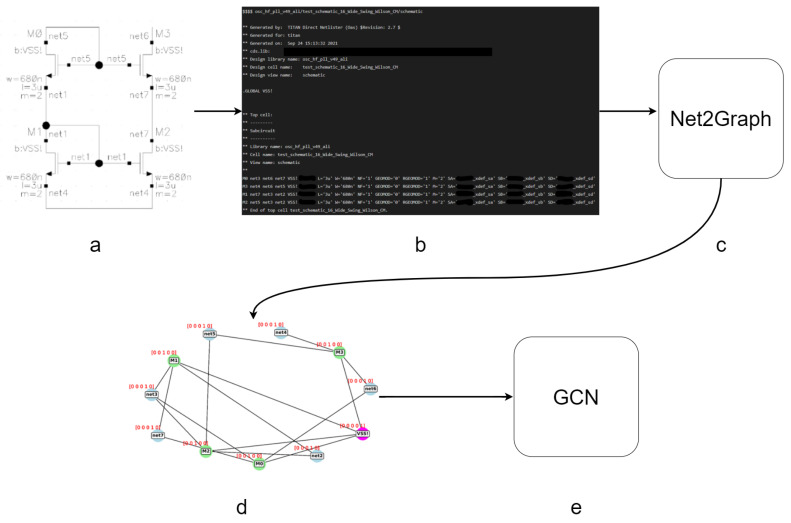
System overview: a pipeline for one sample classification using GCN. (**a**) Input schematic, (**b**) the schematic is converted to Spice netlist using Cadence Virtuoso, (**c**) Spice netlist parser, (**d**) the parser is used to convert the netlist into a graph, (**e**) GCN is the chosen neural network model to perform the classification.

**Figure 6 sensors-23-02989-f006:**
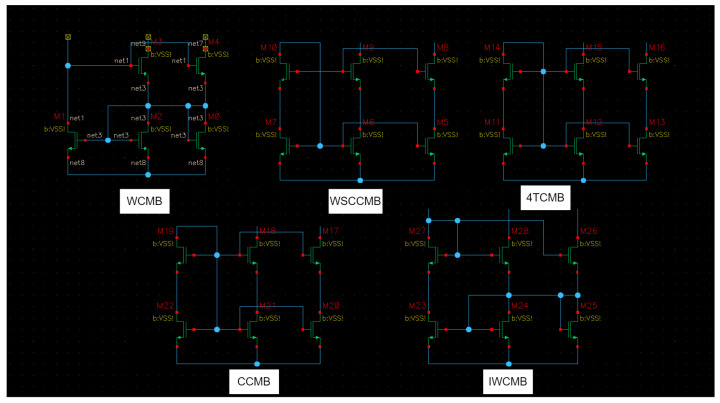
Five classes of current mirror banks to be classified: Cascode Current Mirror Bank(CCMB), 4 Transistor Current Mirror Bank (4TCMB), Wilson Current Mirror Bank (WCMB), Improved Wilson Current Mirror Bank (IWCMB), Wide Swing Cascode Current Mirror Bank (WSCCMB). Each one has an extra bank and the possibility to add more banks on the right side [24].

**Figure 7 sensors-23-02989-f007:**
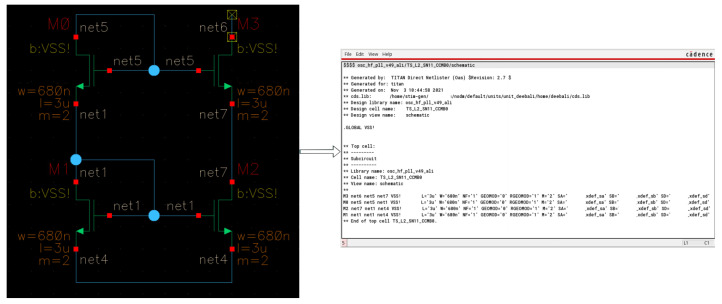
Dataset Transformation: Converting schematics to netlists using Cadence Virtuoso. The connectivity of the transistors is described after each transistor name in text form, followed by the electrical description and characterization of each transistor.

**Figure 8 sensors-23-02989-f008:**
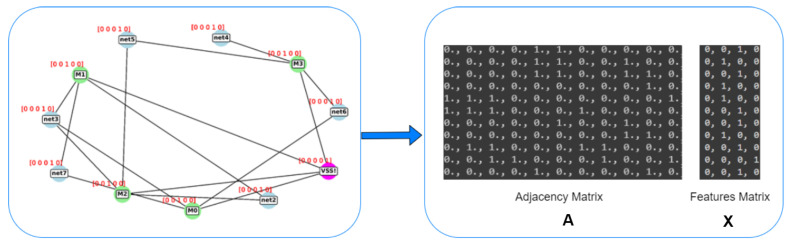
Dataset preprocessing: Converting the netlist to annotated graphs, then representing each graph with adjacency and feature matrices.

**Figure 9 sensors-23-02989-f009:**
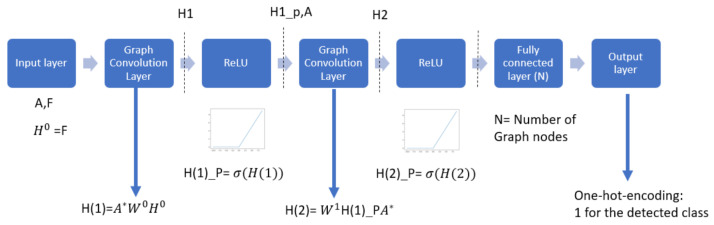
The architecture of the GCN classifier model. Using the fast approximation of spectral convolution, H1 and H2 are the output of the first and second convolution layers, calculated using $A^{*}$, the normalized Adjacency matrix [41], a. ReLU layers remove the negative values, and finally, the SoftMax layer votes for the detected class.

**Figure 10 sensors-23-02989-f010:**
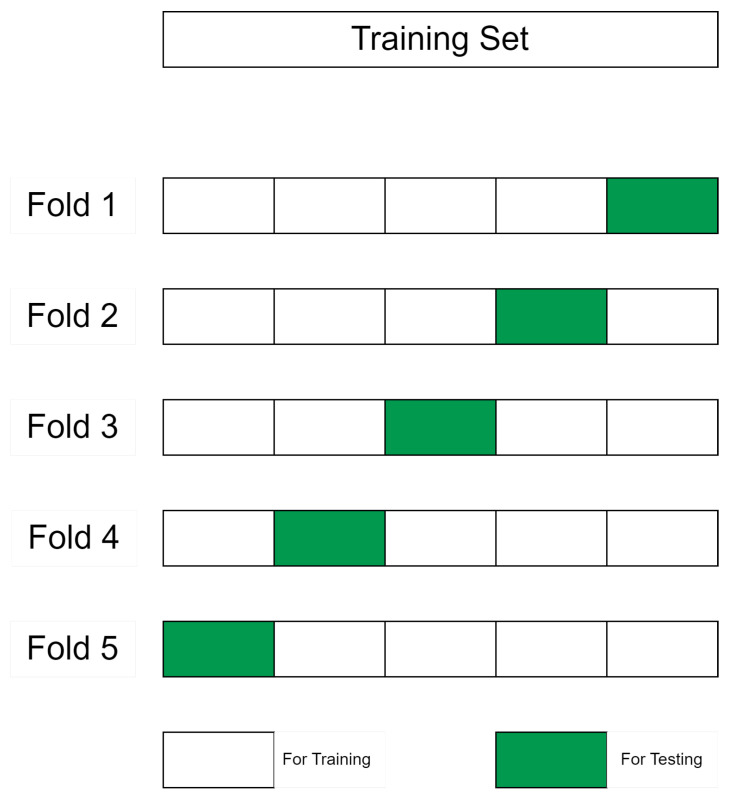
Kfold cross-validation mechanism: in each iteration, a part of the dataset is used for testing the model while the rest is used for training the model independently. In other iterations, another part is used for testing, and so on [56].

**Figure 11 sensors-23-02989-f011:**
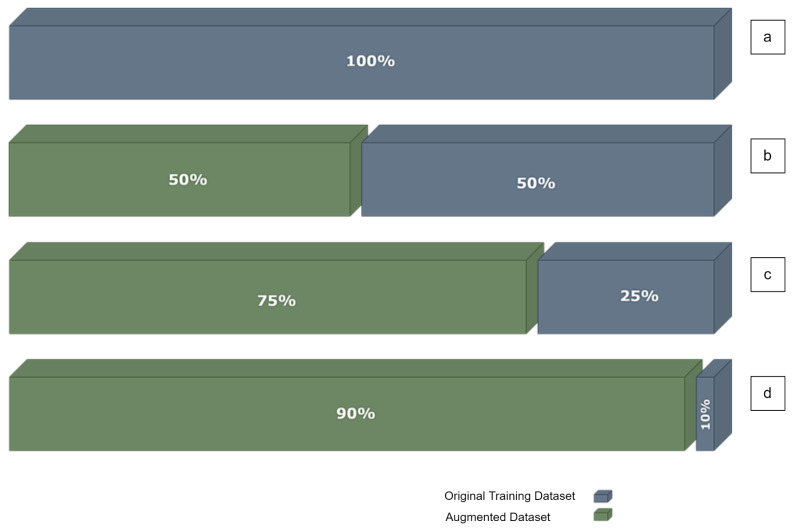
(**a**) Training dataset TDS. (**b**) TDS/2: 50% of the training dataset and 50% augmented data. (**c**) TDS/4: 25% of the training dataset and 75% augmented data. (**d**) TDS/10: 10% of the training dataset and 90% augmented data.

**Figure 12 sensors-23-02989-f012:**
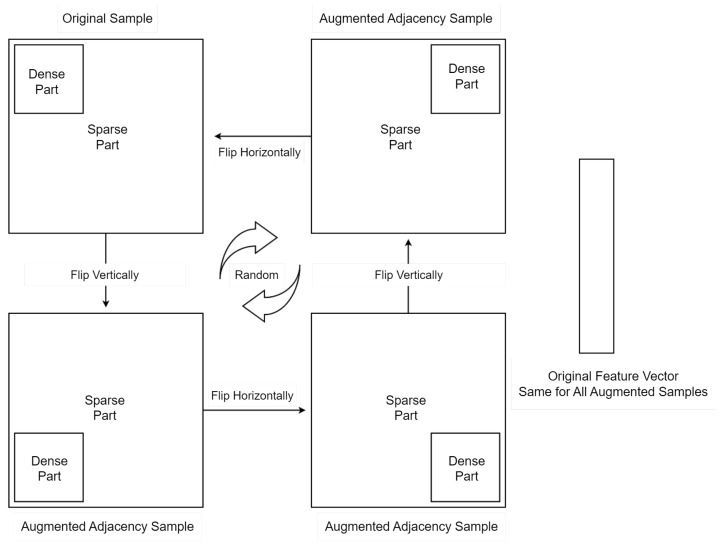
For each adjacency matrix in the training set, the dense part is flipped systematically and the original feature vector is fixed for all augmented samples. Then we extend the training dataset with the new augmented adjacency matrix concatenated with the original feature vector as follows [Adj Augmented | X Original].

**Figure 13 sensors-23-02989-f013:**
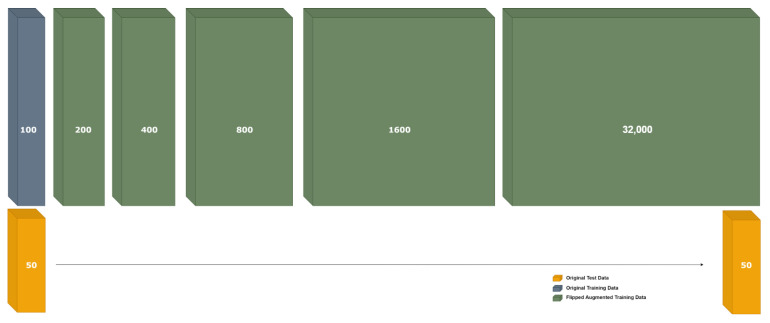
Augmentation by flipping: Original test dataset was kept the same. On the other hand, the training dataset was augmented from 100 graphs up to 32,000 graphs through 5 stages.

**Figure 14 sensors-23-02989-f014:**
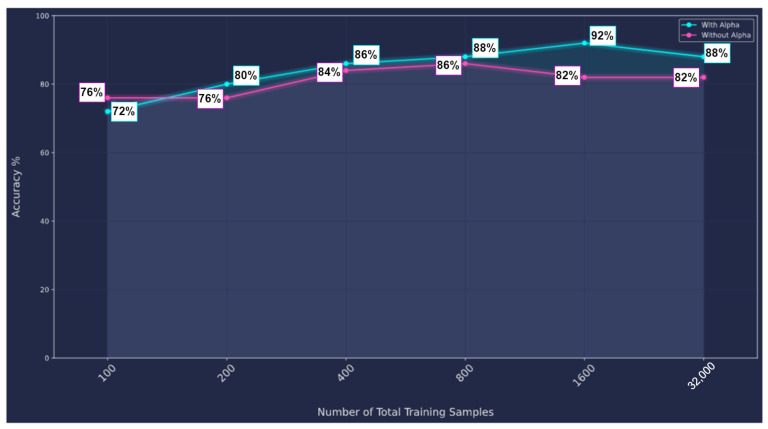
Augmentation by flipping: Original test dataset was kept the same. On the other hand, the training dataset was augmented from 100 graphs up to 32,000 graphs through 5 stages.

**Figure 15 sensors-23-02989-f015:**
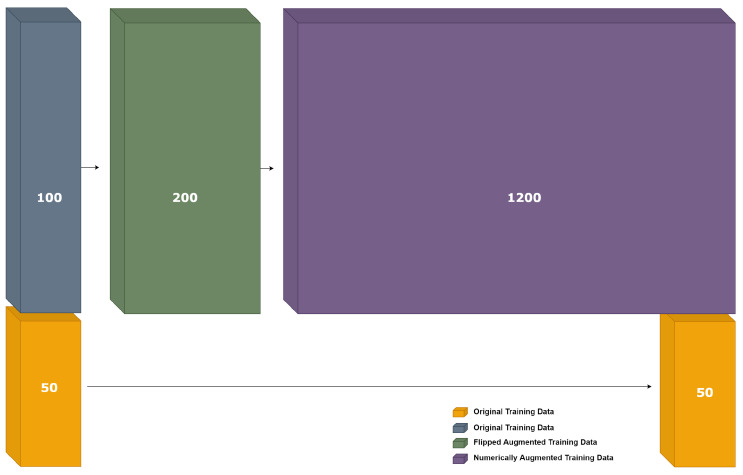
Multistage augmentation visualization: Data augmented first by flipping and then numerically.

**Figure 16 sensors-23-02989-f016:**
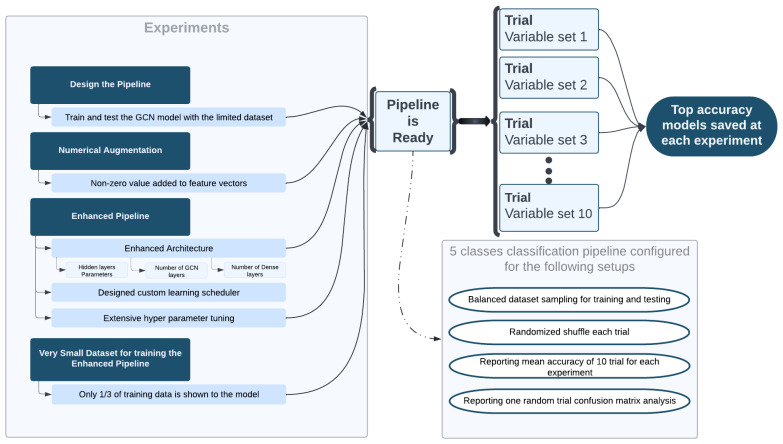
Block diagram of the experiments anatomy of the pipelines for describing the progress until developing the finally used best pipeline.

**Table 1 sensors-23-02989-t001:** An overview of the most relevant related works w.r.t. an assessment of how far they do satisfy the six requirements formulated and outlined in Section 2 (see Requirements Engineering dossier).

Most Relevant Related Works	Classification Accuracy in Range 97–99% (REQ-1)	Robustness w.r.t. Architecture Variations (REQ-2)	Robustness w.r.t. Architecture Extension through Banks (REQ-3)	Robustness w.r.t. Transistor Technologies Variations (REQ.4)	Robustness w.r.t. Training Dataset Imperfections (REQ-5)	Applicability to Higher Levels of AMS IP Stacks
Kunal et al. [4]	Yes	Yes	No	No	No	Yes, possibly
Hong et al. [5]	Yes	Yes	No	No	No	Yes, possibly
Zhang et al. [19]	Yes	Yes	No	No	No	Yes, possibly
Ahmad et al. [24]	Yes	Yes	No	No	No	Yes, possibly

**Table 2 sensors-23-02989-t002:** Created dataset in Cadence Virtuoso: For each class of current mirror banks, 10 schematics were created by adding more banks. The first sample was created with no banks at all, and the last one was created with nine banks.

Wide Swing Cascode Current Mirror Bank	Cascode Current Mirror Bank	4 Transistor Current Mirror Bank	Wilson Current Mirror Bank	Improved Wilson Current Mirror Bank
TS_L2_SN1_WSCCMB0	TS_L2_SN11_CCMB0	TS_L2_SN21_4TCMB0	TS_L2_SN31_WCMB0	TS_L2_SN41_ IWCMB0
TS_L2_SN2_WSCCMB1	TS_L2_SN12_CCMB1	TS_L2_SN22_4TCMB1	TS_L2_SN32_WCMB1	TS_L2_SN42_ IWCMB1
TS_L2_SN3_WSCCMB2	TS_L2_SN13_CCMB2	TS_L2_SN23_4TCMB2	TS_L2_SN33_WCMB2	TS_L2_SN43_ IWCMB2
TS_L2_SN4_WSCCMB3	TS_L2_SN14_CCMB3	TS_L2_SN24_4TCMB3	TS_L2_SN34_WCMB3	TS_L2_SN44_ IWCMB3
TS_L2_SN5_WSCCMB4	TS_L2_SN15_CCMB4	TS_L2_SN25_4TCMB4	TS_L2_SN35_WCMB4	TS_L2_SN45_ IWCMB4
TS_L2_SN6_WSCCMB5	TS_L2_SN16_CCMB5	TS_L2_SN26_4TCMB5	TS_L2_SN36_WCMB5	TS_L2_SN46_ IWCMB5
TS_L2_SN7_WSCCMB6	TS_L2_SN17_CCMB6	TS_L2_SN27_4TCMB6	TS_L2_SN37_ WCMB6	TS_L2_SN47_ IWCMB6
TS_L2_SN8_WSCCMB7	TS_L2_SN18_CCMB7	TS_L2_SN28_4TCMB7	TS_L2_SN38_ WCMB7	TS_L2_SN48_ IWCMB7
TS_L2_SN9_WSCCMB8	TS_L2_SN19_CCMB8	TS_L2_SN29_4TCMB8	TS_L2_SN39_ WCMB8	TS_L2_SN49_ IWCMB8
TS_L2_SN10_WSCCMB9	TS_L2_SN20_CCMB9	TS_L2_SN30_4TCMB9	TS_L2_SN40_ WCMB9	TS_L2_SN50_ IWCMB9

**Table 3 sensors-23-02989-t003:** Five unique types of elements are considered for this classification problem.

Node Type	Feature Vector
NMOS [CMOS]	[0 1 0 0 0]
PMOS [CMOS]	[0 0 1 0 0]
BIPOLAR	[1 0 0 0 0]
N- Voltage Source	[0 0 0 0 1]
N-Net	[0 0 0 1 0]

**Table 4 sensors-23-02989-t004:** K-Fold (5 folds case): 1/5 of the dataset is used for testing and the rest is used for training. A confusion matrix is drawn for each one of the five folds. They show 100% accuracy for fold 4, and 90% accuracy for fold 3.

**Score for fold 1—Accuracy of 100%**						
True labels	4TCM	100% (9/9)				
	WCM		100% (4/4)			
	WSCCM			100% (6/6)		
	IWCM				100% (6/6)	
	CCM					100% (5/5)
		4TCM	WCM	WSCCM	IWCM	CCM
		Predicted labels				
**Score for fold 2—Accuracy of 100%**						
True labels	4TCM	100% (8/8)				
	WCM		100% (9/9)			
	WSCCM			100% (3/3)		
	IWCM				100% (6/6)	
	CCM					100% (4/4)
		4TCM	WCM	WSCCM	IWCM	CCM
		Predicted labels				
**Score for fold 3— Accuracy of 90%**						
True labels	4TCM	100% (3/3)				
	WCM		100% (8/8)			
	WSCCM	30% (3)		100% (7/10)		
	IWCM				100% (2/2)	
	CCM					100% (7/7)
		4TCM	WCM	WSCCM	IWCM	CCM
		Predicted labels				
**Score for fold 4—Accuracy of 100%**						
True labels	4TCM	100% (5/5)				
	WCM		100% (3/3)			
	WSCCM			100% (4/4)		
	IWCM				100% (9/9)	
	CCM					100% (9/9)
		4TCM	WCM	WSCCM	IWCM	CCM
		Predicted labels				
**Score for fold 5—Accuracy of 100%**						
True labels	4TCM	100% (5/5)				
	WCM		100% (6/6)			
	WSCCM			100% (7/7)		
	IWCM				100% (7/7)	
	CCM					100% (5/5)
		4TCM	WCM	WSCCM	IWCM	CCM
		Predicted labels				

**Table 5 sensors-23-02989-t005:** A total of 15 folds were applied on the dataset, which means in each fold 1/15 of the dataset is used for testing and the rest 14/15 is used for training independently. For each fold, a confusion matrix is drawn, and all confusion matrices show 100% accuracy of detection.

**Score for fold 1—Accuracy of 100%**						
True labels	4TCM	100% (4/4)				
	WCM		100% (2/2)			
	WSCCM			100% (2/2)		
	IWCM				100% (2/2)	
		4TCM	WCM	WSCCM	IWCM	
		Predicted labels				
**Score for fold 2—Accuracy of 100%**						
True labels	4TCM	100% (5/5)				
	WCM		100% (2/2)			
	WSCCM			100% (3/3)		
		4TCM	WCM	WSCCM		
		Predicted labels				
**Score for fold 3—Accuracy of 100%**						
True labels	4TCM	100% (1/1)				
	WCM		100% (1/1)			
	WSCCM			100% (6/6)		
	IWCM				100% (2/2)	
		4TCM	WCM	WSCCM	IWCM	
		Predicted labels				
**Score for fold 4—Accuracy of 100%**						
True labels	4TCM	100% (6/6)				
	WCM		100% (4/4)			
		4TCM	WCM			
		Predicted labels				
**Score for fold 5—Accuracy of 100%**						
True labels	4TCM	100% (4/4)				
	WCM		100% (2/2)			
	WSCCM			100% (3/3)		
	IWCM				100% (1/1)	
		4TCM	WCM	WSCCM	IWCM	
		Predicted labels				
**Score for fold 6—Accuracy of 100%**						
True labels	4TCM	100% (3/3)				
	WCM		100% (1/1)			
	WSCCM			100% (2/2)		
	IWCM				100% (2/2)	
	CCM					100% (2/2)
		4TCM	WCM	WSCCM	IWCM	CCM
		Predicted labels				
**Score for fold 7—Accuracy of 100%**						
True labels	4TCM	100% (1/1)				
	WCM		100% (2/2)			
	WSCCM			100% (1/1)		
	IWCM				100% (2/2)	
	CCM					100% (4/4)
		4TCM	WCM	WSCCM	IWCM	CCM
		Predicted labels				
**Score for fold 8—Accuracy of 100%**						
True labels	4TCM	100% (1/1)				
	WCM		100% (1/1)			
	WSCCM			100% (3/3)		
	IWCM				100% (3/3)	
	CCM					100% (2/2)
		4TCM	WCM	WSCCM	IWCM	CCM
		Predicted labels				
**Score for fold 9—Accuracy of 100%**						
True labels	4TCM	100% (4/4)				
	WCM		100% (4/4)			
	WSCCM			100% (2/2)		
		4TCM	WCM	WSCCM		
		Predicted labels				
**Score for fold 10—Accuracy of 100%**						
True labels	4TCM	100% (2/2)				
	WCM		100% (2/2)			
	WSCCM			100% (3/3)		
	IWCM				100% (2/2)	
	CCM					100% (1/1)
		4TCM	WCM	WSCCM	IWCM	CCM
		Predicted labels				
**Score for fold 11—Accuracy of 100%**						
True labels	4TCM	100% (1/1)				
	WCM		100% (1/1)			
	WSCCM			100% (3/3)		
	IWCM				100% (4/4)	
	CCM					100% (1/1)
		4TCM	WCM	WSCCM	IWCM	CCM
		Predicted labels				
**Score for fold 12—Accuracy of 100%**						
True labels	4TCM	100% (4/4)				
	WCM		100% (3/3)			
	WSCCM			100% (3/3)		
		4TCM	WCM	WSCCM		
		Predicted labels				
**Score for fold 13—Accuracy of 100%**						
True labels	4TCM	100% (1/1)				
	WCM		100% (2/2)			
	WSCCM			100% (4/4)		
	IWCM				100% (1/1)	
	CCM					100% (2/2)
		4TCM	WCM	WSCCM	IWCM	CCM
		Predicted labels				
**Score for fold 14—Accuracy of 100%**						
True labels	4TCM	100% (1/1)				
	WCM		100% (2/2)			
	WSCCM			100% (3/3)		
	IWCM				100% (1/1)	
	CCM					100% (3/3)
		4TCM	WCM	WSCCM	IWCM	CCM
		Predicted labels				
**Score for fold 15—Accuracy of 100%**						
True labels	4TCM	100% (1/1)				
	WCM		100% (2/2)			
	WSCCM			100% (6/6)		
	IWCM				100% (1/1)	
		4TCM	WCM	WSCCM	IWCM	
		Predicted labels				

**Table 11 sensors-23-02989-t011:** (**a**). Confusion matrix for test using 50 graphs, where the GCN model was trained using 100 graphs. (**b**). Confusion matrix after a “multistage augmentation”: the confusion matrix shows a 100% classification accuracy.

(**a**)						
True	4TCM	100% (10/10)				
	WCM		100% (10/10)			
	WSCCM			100% (10/10)		
	IWCM				100% (10/10)	
	CCM	10%(1/10)				100% (9/10)
		4TCM	WCM	WSCCM	IWCM	CCM
		Predicted				
(**b**)						
True	4TCM	100% (10/10)				
	WCM		100% (10/10)			
	WSCCM			100% (10/10)		
	IWCM				100% (10/10)	
	CCM					100% (10/10)
		4TCM	WCM	WSCCM	IWCM	CCM
		Predicted				

**Table 12 sensors-23-02989-t012:** Number of features per class while before applying the hyperphysical augmentation.

Feature Type	Number of Features
NMOS [CMOS]	1
PMOS [CMOS]	1
Bipolar	1
N-Net	1

**Table 13 sensors-23-02989-t013:** Number of features per class while taking into consideration the hyperphysical augmentation.

Feature Type	Number of Features
NMOS [CMOS]	4
PMOS [CMOS]	4
Bipolar	4
N-Net	1

**Table 14 sensors-23-02989-t014:** Thirteen unique types of elements are considered for Features Encoding.

Node Type	Feature Vector
NMOS [CMOS]	[1 0 0 0 0 0 0 0 0 0 0 0 0]
NMOS [CMOS][1]	[0 1 0 0 0 0 0 0 0 0 0 0 0]
NMOS [CMOS][2]	[0 0 1 0 0 0 0 0 0 0 0 0 0]
NMOS [CMOS][3]	[0 0 0 1 0 0 0 0 0 0 0 0 0]
PMOS [CMOS]	[0 0 0 0 1 0 0 0 0 0 0 0 0]
PMOS [CMOS][3]	[0 0 0 0 0 1 0 0 0 0 0 0 0]
PMOS [CMOS][2]	[0 0 0 0 0 0 1 0 0 0 0 0 0]
PMOS [CMOS][3]	[0 0 0 0 0 0 0 1 0 0 0 0 0]
BIPOLAR	[0 0 0 0 0 0 0 0 1 0 0 0 0]
BIPOLAR[1]	[0 0 0 0 0 0 0 0 0 1 0 0 0]
BIPOLAR[2]	[0 0 0 0 0 0 0 0 0 0 1 0 0]
BIPOLAR[3]	[0 0 0 0 0 0 0 0 0 0 0 1 0]
N-Net	[0 0 0 0 0 0 0 0 0 0 0 0 1]

**Table 15 sensors-23-02989-t015:** Five classes count after performing “Hyperphysical Augmentation” for increasing the number of samples from 150 to 600, and thus 120 samples per class.

Class	Count
Cascode Current Mirror Bank (CCMB)	120
4 Transistor Current Mirror Bank (4TCMB)	120
Wilson Current Mirror Bank (WCMB)	
Improved Wilson Current Mirror Bank (IWCMB)	120
Wide Swing Cascode Current Mirror Bank (WSCCMB)	120

**Table 16 sensors-23-02989-t016:** Confusion matrix showing the performance after a “Hyperphysical Augmentation” of limited dataset size. The reached result: 100% accuracy.

True	4TCM	100% (40/40)				
	WCM		100% (40/40)			
	WSCCM			100% (40/40)		
	IWCM				100% (40/40)	
	CCM					100% (40/40)
		4TCM	WCM	WSCCM	IWCM	CCM
		Predicted				

**Table 17 sensors-23-02989-t017:** Performance comparison; TDS is the dataset where 400 graphs are for training and 200 graphs for testing. The Numerical Augmentation ratio is Dataset × Augmentation-Scale.

Dataset	TDS	TDS/2	TDS/4	TDS/10
Mean Accuracy	100.00 ± 0.00	98.20 ± 0.32	69.47 ± 0.57	49.85 ± 0.34
Augmented Dataset	TDS	(TDS/2)*2	(TDS/4)*4	(TDS/10)*10
Mean Accuracy	100.00 ± 0.00	100.00 ± 0.00	92.15 ± 0.46	61.55 ± 0.50

**Table 18 sensors-23-02989-t018:** Numerical augmentation of the hyperphysically augmented dataset: this confusion matrix shows that all classes were classified correctly and this results in a 100% accuracy.

True	4TCM	100% (40/40)				
	WCM		100% (40/40)			
	WSCCM			100% (40/40)		
	IWCM				100% (40/40)	
	CCM					100% (40/40)
		4TCM	WCM	WSCCM	IWCM	CCM
		Predicted				

**Table 19 sensors-23-02989-t019:** A comprehensive summary of how far all elements of the comprehensive requirements dossier have been fully satisfied, and a brief comparison to the respective related works.

REQ-ID from the Requirements Engineering Dossier	What This Paper Has Specifically Done w.r.t. to REQ-ID	A Brief Commenting of a Sample of Relevant Related Work(s)	General Comments
REQ-1	In this work, we managed to reach 100% accuracy of classification	Refs. [4,5,19,24] achieved above 98% accuracy of classification	our paper is as good as the relevant related works w.r.t. REQ-1
REQ-2	Our model is robust w.r.t. topology variations of circuits having the same label.	The model was robust against differences in circuit topology and was able to accurately identify the circuits regardless of the variations in their structure that occurred within a single class, which was achieved by [4,5] in circuits, and in [19,24] in graph domain.	This paper does as well as the relevant related works w.r.t. REQ-2
REQ-3	robust w.r.t. to topology variations/extension related to banks (e.g., current mirror banks of different dimensions; etc.)	Refs. [4,5,19,24] did not discuss structure extension	This paper outperforms the relevant related works w.r.t. REQ-3
REQ-4	Our model was not sensitive to transistor technology changes in dataset samples classified as the same class (NMOS, PMOS, or Bipolar).	This was not the case in [4,5], and it is not valid in [19] nor in [24] since they classify graphs.	This paper outperform the relevant related works w.r.t. REQ-4
REQ-5	The most recent research findings were presented, each of which was derived from a larger dataset than the one we used.	In [4], they utilized a dataset with 1232 circuits, but we used a dataset with just 50 circuits. In [5], 80 graphs were used in training and 144 for testing. Regarding graph classification, it is easier to create a huge number of graphs which is not the case in schematics. For example, 50,000 graphs were used in [19] for multi-class classification.	This paper outperforms the relevant related works w.r.t. REQ-5
REQ-6	This paper presented a proof of concept, and the pipeline is capable of classifying any other higher level should the information be provided	In [4,5] circuits of the same level were classified, and the scalability to higher hierarchies of the issue of the circuit was not mentioned. In [19,24], graphs of the same size were classified as well hierarchal graphs were not discussed.	This paper outperforms the relevant related works w.r.t. REQ-6

## Data Availability

All the data have been included in the study.

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
