# Peer review of "A Robust Automated Analog Circuits Classification Involving a Graph Neural Network and a Novel Data Augmentation Strategy"

_sensors, 2023, doi:10.3390/s23062989_

Round 1
Reviewer 1 Report
This paper discusses the need for automation of analog circuit design and verification processes due to their time-consuming nature and reliance on human experience. The authors aim to develop a robust and reliable analog circuit classification system using graph neural networks (GCN) and novel dataset augmentation techniques. The approach involves converting the circuit's schematics into graph representations and using a GCN processor to determine the label of the input analog circuit. The notable contributions of the paper include a comprehensive review of the state-of-the-art in analog circuit classification, a GCN model for analog circuit classification, and a novel augmentation strategy to overcome limitations in training datasets. The authors will perform comprehensive benchmarking to compare the performance of their proposed solution with existing related works.
The idea of using graph representations of circuits and GCN for circuit classification is interesting and has potential in the field of AMS verification. The novelty of the data augmentation strategy is also relevant for the scientific comunity. Overall, the paper presents a promising approach for automatic classification of analog circuits.
I have enjoyed the reading.
My suggestions and comments follows:
- Beyond the idea of automation of analog circuit design and verification processes, there is also the "Design cycle" of rapid prototyping from the concept to the Silicon (prototype). do you think that introducing a design platform aimed at speeding up the process of bringing a project from concept to tapeout is also add value to the design automation pipeline. Note: its not mandatory to include this comment in the revised manuscript. The Reviewer is asking for a professional opinion.
- Sec. 6, par. 1: "CNNs do operate on neighboring pixels (structured data) and have reached great results in applications such as image classification, object detection, semantic segmentation, etc.)". more refs that exhibit the listed applications should be given. There are a bunch of interesting works that show analog image classification. For instance:
[leek2022deep] Leek, E. Charles, Ales Leonardis, and Dietmar Heinke. "Deep neural networks and image classification in biological vision." Vision Research 197 (2022): 108058.
[zambrano2022all] Zambrano, Benjamin, et al. "All-Analog Silicon Integration of Image Sensor and Neural Computing Engine for Image Classification." IEEE Access 10 (2022): 94417-94430.
[cao2022applications] Cao, Pingping, et al. "Applications of graph convolutional networks in computer vision." Neural Computing and Applications 34.16 (2022): 13387-13405.
- What about "Graph pre-processing"? improving the quality of the graph representation of the analog circuits can significantly impact the performance of GCN.
- I believe a comparison against other GCN is missing. can you comment on different GCN architectures as compared to the proposed one? If it cannot be done, at least a qualitative comparison mentioning the advantages and disadvantages.
- The quality of all the figures MUST be improved. There are some figures in which a frame is seen at the corners.
- line 615: authors forgot a reference (typo)
- proofread you manuscript
Author Response
Responses to Reviewer 1’s Comments
Manuscript ID: sensors-2225350
Point 1: Beyond the idea of automation of analog circuit design and verification processes, there is also the "Design cycle" of rapid prototyping from the concept to the Silicon (prototype). do you think that introducing a design platform aimed at speeding up the process of bringing a project from concept to tapeout is also add value to the design automation pipeline. Note: its not mandatory to include this comment in the revised manuscript. The Reviewer is asking for a professional opinion.
Response to Point 1:
In the current work, we do target to contribute to shortening the development cycle (time to market) for the cases where the circuitry (analog design) is already available or several samples have already been designed. Design, probably, was already done by experts (or is ongoing), and we try to analyze different bricks in order to identify known structures to automate and assist in later steps of the IC development flow. We assume the concept problems are to be sorted out, and we only try to assist with steps mostly after (but eventually also during) the analog design phase. Rapid prototyping is a powerful approach that allows the testing of various concepts quickly. But the concept step is outside the scope of this research. Thus, it was not considered. A domain expert at Infineon Technologies (Austria) has confirmed this.
Point 2: Sec. 6, par. 1: "CNNs do operate on neighboring pixels (structured data) and have reached great results in applications such as image classification, object detection, semantic segmentation, etc.)". more refs that exhibit the listed applications should be given. There are a bunch of interesting works that show analog image classification. For instance:
[leek2022deep] Leek, E. Charles, Ales Leonardis, and Dietmar Heinke. "Deep neural networks and image classification in biological vision." Vision Research 197 (2022): 108058.
[zambrano2022all] Zambrano, Benjamin, et al. "All-Analog Silicon Integration of Image Sensor and Neural Computing Engine for Image Classification." IEEE Access 10 (2022): 94417-94430.
[cao2022applications] Cao, Pingping, et al. "Applications of graph convolutional networks in computer vision." Neural Computing and Applications 34.16 (2022): 13387-13405.
Response to Point 2:
In lines [532-536] in the newly revised manuscript, the three references are now cited as follows:
<<For instance, feedforward deep convolutional neural networks (fDCNNs) were used to classify images in biological system [50]. Moreover, in [51], an analog artificial neural network was used for low-resolution CMOS image classification. Finally, a review of GCN applications in computer vision, such as object detection, image recognition, object tracking, and instance segmentation, was shown in [52]. >>
Point 3: What about "Graph pre-processing"? improving the quality of the graph representation of the analog circuits can significantly impact the performance of GCN.
Response to Point 3:
In the data domain, graph preprocessing is an essential step. For example, in image classification, it is highly recommended to improve the input of the GCN or the CNN, as shown in [r3][ r4][ r5]. On the other hand, the input of the GCN does not need preprocessing, as proved in [10]. Indeed, Ref. [10] shows that preprocessing, if needed, can be integrated in the graph neural network structure.
Moreover, in our case, we have not needed graph preprocessing, which even eventually would alter the equivalent circuit and affect its functionality. Hence, have not included any preprocessing in our model.
Point 4: I believe a comparison against other GCN is missing. can you comment on different GCN architectures as compared to the proposed one? If it cannot be done, at least a qualitative comparison mentioning the advantages and disadvantages.
Response to Point 4:
In lines [571-572], the following sentence has been added:
<< Empirically, it has been found that two-graph-convolution-layers GCN work the best, and the decision was taken depending on the grid search mechanism for finding the best-performing structure.>>
In lines [539-580] of the newly revised manuscript, we have shown why we chose spectral GCN instead of spatial GCN. We have chosen the Vanilla GCN, which uses an approximate convolution on graphs that is equivalent to spectral graph convolution [46]. It is the simplest type of GCN and does perform very well on classification applications, as we have shown in this paper. The other type of GCN, the spatial GCN, is mainly used for other tasks, such as graph clustering since it focuses on the local structures of the graph[r1][r2]. There exist further GCN models, for example, the ones including some attention mechanism. We did not consider the inclusion of attention mechanisms, which is a very advanced functionality, as our current model was already robust enough to reach 100% accuracy through dataset augmentations.
Regarding the GCN structures, we have used different configurations and tried many simulations to decide what structure to use. In [1], it was recommended to use a two-layer structure. Nevertherless, we have explored the performance of the following structure configurations:
- One-layer graph-convolution GCN architecture
- Two-layer graph-convolution GCN architecture
- Three-layer graph-convolution GCN architecture
- Four-layer graph-convolution GCN architecture
- Five-layer graph-convolution GCN architecture
Moreover, we have also tried parallel structures with 3*3 graph convolution layers followed by another graph convolution layer, as shown in Figure 1.
The performance results obtained for all structures were compared, and the two-layer graph-convolution GCN architecture performed at best, as well as the parallel structure shown in Figure 1. Therefore, the two-layer architecture presented in Figure 9 of the manuscript was chosen as it does have a much-lower complexity than parallel architectures at comparable performance.
Figure 1 parallel GCN structure: 3*3 graph convolution layers followed by another graph convolution layer, and finally the fully connected part.
Point 5: The quality of all the figures MUST be improved. There are some figures in which a frame is seen at the corners.
Response to Point 5:
All figures were redone, and the frame issue has been solved in the new manuscript version.
Point 6: line 615: authors forgot a reference (typo)
Response to Point 6: This issue is addressed in the new manuscript.
Point 7: proofread your manuscript
Response to Point 7: This issue has been carefully addressed in the new manuscript version.
References as numbered in the manuscript:
[1] Kunal, Kishor, et al. "GANA: Graph convolutional network based automated netlist annotation for analog circuits." 2020 Design, Automation & Test in Europe Conference & Exhibition (DATE). IEEE, 2020
[10] Zhang, Muhan, et al. "An end-to-end deep learning architecture for graph classification." Proceedings of the AAAI conference on artificial intelligence. Vol. 32. No. 1. 2018.
[46] T. N. Kipf and M. Welling, “Semi-supervised classification with graph convolutional networks,” arXiv [cs.LG], 2016.
[50] Leek, E. Charles, Ales Leonardis, and Dietmar Heinke. "Deep neural networks and image classification in biological vision." Vision Research 197 (2022): 108058.
[51] B. Zambrano, S. Strangio, T. Rizzo, E. Garzón, M. Lanuzza and G. Iannaccone, "All-Analog Silicon Integration of Image Sensor and Neural Computing Engine for Image Classification," in IEEE Access, vol. 10, pp. 94417-94430, 2022, doi: 10.1109/AC-CESS.2022.3203394.
[52] Cao, P., Zhu, Z., Wang, Z. et al. Applications of graph convolutional networks in computer vision. Neural Comput Applic 34,13387–13405 (2022). https://doi.org/10.1007/s00521-022-07368-1
New References:
[r1] K. Settaluri and E. Fallon, “Fully Automated Analog subCircuit Clustering with Graph Convolutional Neural Networks,” in 2020 Design, Automation & Test in Europe Conference & Exhibition (DATE), 2020, pp. 1714–1715. doi: 10.23919/DATE48585.2020.9116513.
[r2] R. Patel, H. Habal, and K. Venkata, “Machine Learning based Structure Recognition in Analog Schematics for Constraints Generation,” DVCON Design and Verification, 2021.
[r3] Tavakkoli, Vahid, Kabeh Mohsenzadegan, and Kyandoghere Kyamakya. "A visual sensing concept for robustly classifying house types through a convolutional neural network architecture involving a multi-channel features extraction." Sensors 20.19 (2020): 5672.
[r4] Mohsenzadegan, Kabeh, Vahid Tavakkoli, and Kyandoghere Kyamakya. "A Deep-Learning Based Visual Sensing Concept for a Robust Classification of Document Images under Real-World Hard Conditions." Sensors 21.20 (2021): 6763.
[r5] Mohsenzadegan, Kabeh, Vahid Tavakkoli, and Kyandoghere Kyamakya. "A Smart Visual Sensing Concept Involving Deep Learning for a Robust Optical Character Recognition under Hard Real-World Conditions." Sensors 22.16 (2022): 6025.

Reviewer 2 Report
The paper deals with A Robust Automated Analog Circuits Classification involving a Graph Neural Network and a Novel Data Augmentation Strategy. The following comments should be addressed to improve the technical quality of the manuscript.
1. Provide quantitative results in the Abstract.
2. Too many long paragraphs, making it difficult to read and comprehend. Break it down into smaller paragraph.
3. Elaborate more on why is it possible to add more banks on the right side of each one of the current mirror banks?
4. At times this paper reads like a thesis. The authors seem to have included all their results, with the consequence that I am not sure which findings are significant and which are not.
5. How do The various augmentation methods specifically alter the graph’s structure?
Author Response
Responses to Reviewer 2’s Comments
Manuscript ID: sensors-2225350
Point 1: Provide quantitative results in the Abstract.
Response to Point 1: We have added some lines describing the quantitative results in the new abstract.
Point 2: Too many long paragraphs, making it difficult to read and comprehend. Break it down into smaller paragraph.
Response to Point 2: This issue has been solved in the revised manuscript’s version.
Point 3. Elaborate more on why is it possible to add more banks on the right side of each one of the current mirror banks?
Response to Point 3: It is well known from the fundamentals of electronics that typically, all analog designers put the reference current on the left side and add the needed banks as many replicas as they need right of that reference current. Nevertheless, it is still possible to add banks on both sides of the reference current branch while maintaining the correct connections of the added transistors. This has been confirmed by some domain experts within Infineon technologies (Austria).
Adding banks, either left or right, in this paper, it does allow us to have diverse structures under the same label, whereby the classifier to be developed in this paper should be robust against those structure variations within the same class or label. In [r1], it is shown that the reference branch is always on the left side and the banks are on the right side.
Point 4. At times this paper reads like a thesis. The authors seem to have included all their results, with the consequence that I am not sure which findings are significant and which are not.
Response to Point 4:
We partly disagree with this reviewer’s comment. The feeling that the paper reads like a thesis is rather a proof that it is very comprehensive.
Given the complexity of the core problem to be solved, and in view of the very tough specification book, it is very important to present a comprehensive text which addresses all issues of relevance.
This starts with a comprehensive problem formulation. Then a convincing gap analysis supported by a thorough critical state-of-the-art review. This followed by a comprehensive description of our methodologies and our novel concepts suggested. To finish with a structured presentation of the results obtained, accompanied with a qualitative comparison with competing concepts from the relevant state-of-the-art. It is only through a comprehensive global structure that a complex paper like this one can effectively convince the reader and better highlight the essential contributions of our work presented herein. It shall also be indicated that we did not include all results, but we have selected and presented only the most essential ones.
Point 5. How do The various augmentation methods specifically alter the graph’s structure?
Response to Point 5:
We have used/developed the following different Data Augmentation strategies:
- Electrical Augmentation: This does not alter the new graph structure.
- Each new augmented sample has only a different feature matrix, but the graph’s structure remains unchanged.
- Feature matrix Augmentation (aka Numerical Augmentation):
- This too does not alter the original graph structure.
- Each new augmented graph sample is represented by the same adjacency matrix, but a random small number is added to the non-zero elements of the feature matrix.
- Dataset Augmentation by Flipping:
- This does not alter the new graph structure.
- The adjacency matrix is moved into a bigger sparse matrix to create new augmented samples. Nevertheless, the structure remains the same since we are not altering the graph’s connectivity by keeping the original adjacency matrix untouched.
- Multi-Stage Augmentation:
- The components of this augmentation are flipping + numerical. Hence the is no change in the original graph’s structure.
- Hyperphysical Augmentation:
- This strategy is very similar case to the so-called electrical augmentation.
- The features are represented differently, as described in lines [861-901], but the graph structure remains unchanged.
New References
[r1] Razavi, Behzad. Design of analog CMOS integrated circuits. 2005.[135-144]

Round 2
Reviewer 1 Report
The authors have addressed all the concerns. I’m satisfied with the responses.